# Structured connectivity in the output of the cerebellar cortex

Kim M. Gruver [1,2,5], Jenny W. Y. Jiao [1], Eviatar Fields [1,2], Sen Song[3], Per Jesper Sjöström [4] & Alanna J. Watt [1] ✉

The spatial organization of a neuronal circuit is critically important for its function since the location of neurons is often associated with function. In the cerebellum, the major output of the cerebellar cortex are synapses made from Purkinje cells onto neurons in the cerebellar nuclei, yet little has been known about the spatial organization of these synapses. We explored this question using whole-cell electrophysiology and optogenetics in acute sagittal cerebellar slices to produce spatial connectivity maps of cerebellar cortical output in mice. We observed non-random connectivity where Purkinje cell inputs clustered in cerebellar transverse zones: while many nuclear neurons received inputs from a single zone, several multi-zonal connectivity motifs were also observed. Single neurons receiving input from all four zones were over-represented in our data. These findings reveal that the output of the cerebellar cortex is spatially structured and represents a locus for multimodal integration in the cerebellum.

Connections between neurons give rise to circuits that shape brain function. Identifying how these connections are organized is therefore necessary to understand how a circuit functions. The cerebellum is a highly conserved structure that receives input from many brain regions. In the cerebellar cortex, Purkinje cells are organized into lobules, with both neuroimaging studies and disease research suggesting that Purkinje cells in different regions of the cerebellar cortex process different types of information[1–4].

As the sole output of the cerebellar cortex, Purkinje cell axons target cerebellar nuclear (CN) neurons with strong inhibitory synapses. It is these CN neurons that form the vast majority of output neurons from the cerebellum. Previous studies have suggested that individual CN neurons receive input from ~40 converging Purkinje cells[5], and anatomical studies have shown that narrow bands of neighboring Purkinje cells converge on similar areas in the CN[6,7]. Despite the critical role that the Purkinje cell−CN neuron synapse plays in cerebellar information processing, a functional understanding of how Purkinje cells spatially converge on CN neurons has been lacking.

Previous studies have explored the anatomical properties of Purkinje cell−CN neuron convergence which has been defined longitudinally in the mediolateral plane of the cerebellum as stripes[8–10]. These longitudinal stripes are typically 100−300 μm wide and include Purkinje cells which converge upon CN neurons which themselves project to similar locations within the inferior olive[7], thereby constituting a closed Purkinje cell − CN − inferior olive − Purkinje cell loop[11,12]. In addition to its mediolateral organization into longitudinal stripes, the cerebellar cortex can be further subdivided into four functional transverse zones[13,14] that have been associated with different patterns of gene expression[15,16] and physiological properties[17,18]. These zones include the anterior (lobules II-V), central (VI-VII), posterior (VIII and dorsal IX), and nodular zones (ventral IX and X).

We wanted to explore how Purkinje cell inputs onto CN neurons are spatially organized within the sagittal plane. We focused on the cerebellar fastigial nucleus, since neurons in the fastigial nucleus receive most Purkinje cell input from the vermis[19,20] and are thus preserved within an acute slice preparation. We used electrophysiology and focal optogenetic stimulation to build spatial connectivity maps of Purkinje cell inputs onto individual CN neurons within the sagittal plane. We found that Purkinje cell inputs to the CN are organized along the same four transverse zones that are defined during cerebellar

[1]Department of Biology, McGill University, Montréal, QC, Canada. [2]Integrated Program in Neuroscience, McGill University, Montréal, QC, Canada. [3]Laboratory of Brain and Intelligence and Department of Biomedical Engineering, Tsinghua University, Beijing, China. [4]Department of Neurology and Neurosurgery, McGill University, Montréal, QC, Canada. [5]Present address: Allen Institute for Brain Science, Seattle, WA, USA. ✉e-mail: alanna.watt@mcgill.ca

development[15]. While many CN neurons receive input from a single zone, others receive Purkinje cell input from multiple zones, and connectivity between Purkinje cells and CN neurons is not random. We performed viral labeling of Purkinje cells from different zones and observed that their axons terminate in close proximity within the CN, providing anatomical support for our functional data. Finally, we performed both cell-attached and whole-cell recordings from single CN neurons to determine how Purkinje cell synaptic inputs affect CN output. We found that even small synaptic inputs could shape CN neuron output, suggesting that inputs may not need to synchronize to influence CN output. Our findings support CN neurons as a locus of integration within the cerebellum that is important for cerebellar information processing.

## Results

### Mapping Purkinje cell input onto CN neurons

To build functional connectivity maps of Purkinje-cell convergence onto CN neurons, we used transgenic mice that expressed modified Channelrhodopsin-2 in Purkinje cells, (Fig. 1a–c)[21]. To determine the spread of the light in the tissue, we made control recordings from Purkinje cells' stimulated axons both in the same and adjacent lobules and found that stimulation of lobules adjacent to the patched Purkinje cell did not evoke Purkinje cell activity (Supplementary Fig. 1, 14/15 adjacent lobules tested, with one spike elicited when stimulating a neighboring lobule), consistent with our previous findings[22]. Having confirmed the spatial restriction of our stimulation, we then recorded from CN neurons in voltage-clamp configuration and used focused blue light to stimulate Purkinje cell axons at the base of individual lobules sequentially from lobule II to X (Fig. 1d) to evoke inhibitory postsynaptic currents (IPSCs) in CN neurons. In CN neurons, stimulation of individual lobules often failed to evoke a synaptic response (e.g., Fig. 1d, bottom, where 6 out of 7 stimulated lobules failed to elicit a response), but occasionally IPSCs with fast rise and decay time constants and low failure rates were observed (Fig. 1d, bottom, an IPSC was elicited every time lobule VI/VII was stimulated; Supplementary Fig. 2), consistent with previous descriptions of Purkinje cell – CN synapses on predominantly excitatory neurons in the fastigial nucleus[21,23]. By sequentially stimulating individual lobules with focally restricted light, we identified the location of functional connections between Purkinje cells and CN neurons which enabled us to build connectivity maps of Purkinje cell convergence on individual CN neurons (Fig. 1d, e). As a positive control, we shone light over the Purkinje cell axonal terminals directly surrounding the patched CN neuron soma and were able to produce large IPSCs (Fig. 1d, Axonal terminals; Supplementary Fig. 2). We found that approximately half of CN neurons received no synaptic input from lobule stimulation within the slice (Supplementary Fig. 3), suggesting that they receive input from Purkinje cells outside the zone of stimulation, and likely receive input from Purkinje cells adjacent to the slice.

Determining whether Purkinje cells from a single lobule or multiple lobules (Fig. 1f) converge on an individual CN neuron has important implications for cerebellar function. We found that ~50% of CN neurons that produced IPSCs following lobule stimulation received input from Purkinje cells located in a single lobule (Fig. 1g), while the remaining ~50% of CN neurons received Purkinje cell input from between two and five lobules (Fig. 1h). We observed no differences in IPSC amplitudes from single-lobule stimulation in CN neurons compared with those receiving multi-lobule input (Fig. 1i; Supplementary Fig. 2), suggesting that the total number of spatially distinct inputs does not influence the strength of connections from a single lobule. Furthermore, the cumulative IPSC amplitude for the total input across lobules increased with the number of lobules from which a cell received input (Fig. 1j). To address whether there were differences in the subcellular location of Purkinje cell inputs onto CN neurons, we examined the disparity index[24] of both IPSC amplitudes and rise times.

We found no differences in the properties of IPSCs for individual CN neurons based on the number of connected lobules (Supplementary Fig. 2). These findings suggest that both single-lobule and multi-lobule innervation patterns are observed in Purkinje cell convergence on CN neurons.

Anatomically defined lobules of the cerebellar cortex are not discrete processing units[1,13]. Rather, the cerebellar cortex has been subdivided into four major transverse functional zones within the sagittal plane: the anterior, central, posterior, and nodular zones (Fig. 2a). Since retrograde labeling experiments have shown that CN neurons receive input from narrow bands of neighboring Purkinje cells[6], we wondered whether any patterns existed among inputs from different lobules converging on a CN neuron. To address this, we performed unsupervised hierarchical clustering analysis of our connectivity data. We observed that Purkinje cells in lobules within the same functional zone converged more often onto the same CN neuron than those in lobules from across zones (Fig. 2b). This suggests that Purkinje cell output tends to respect the boundaries of the functional zones of the cerebellar cortex, giving additional evidence for the functional significance of transverse zones in cerebellar information coding.

### Connectivity motifs in Purkinje cell input onto CN neurons

We next chose to look more closely at patterns of connectivity from the perspective of transverse zones and observed both single-zone and multi-zone convergence (Fig. 2c). We observed no differences in IPSC amplitudes or other properties between CN neurons innervated by different zones or by different numbers of zones (Fig. 2d; Supplementary Fig. 2), suggesting that there is no zone that forms a systematically stronger input onto CN neurons. We found that multi-zonal input was not rare: over a third of CNs received multi-zone input (Fig. 2e, inset). Different connectivity patterns appear with different frequencies in our dataset (Supplementary Fig. 4), suggesting the existence of non-random connectivity motifs.

To determine whether Purkinje cells converge across zones on individual CN neurons in non-random motifs[25,26], we used a random model that assumes Purkinje cell connectivity from each zone occurs independently of other zones and determined how well it describes our data (Fig. 2e). We found that 4-zone convergence occurred in our dataset ~18 times more often than predicted by the model, and three-zone connections trended toward significant over-representation as well. This over-representation suggests that a 4-zone connectivity motif may be an important feature of cerebellar information transfer, and by extension, that the CN may be an important locus of information integration in the cerebellum.

### No morphological signature for CN neurons with different input patterns

Different classes of CN neurons have been distinguished by their morphological and physiological properties[27–30]. To determine whether differences in morphological properties can be used to classify CN neurons with different connectivity patterns, we filled CN neurons with AlexaFluor 594 and performed two-photon imaging of filled cells after characterizing their synaptic inputs (Fig. 3a). To characterize their physiological properties, we sampled their spontaneous firing in cell-attached mode prior to recording synaptic input (Fig. 3a, bottom; Supplementary Fig. 5). We compared the soma area, Sholl radius, branch index, number of primary dendrites, total dendrite length, and firing rate from CN neurons with different connectivity patterns and found that these properties were indistinguishable for CN neurons with different connectivity patterns (Fig. 3b–g) or number of input zones (Supplementary Fig. 6). Thus, we observe that CN neurons with different types of input could not be discerned by their morphological or physiological properties.

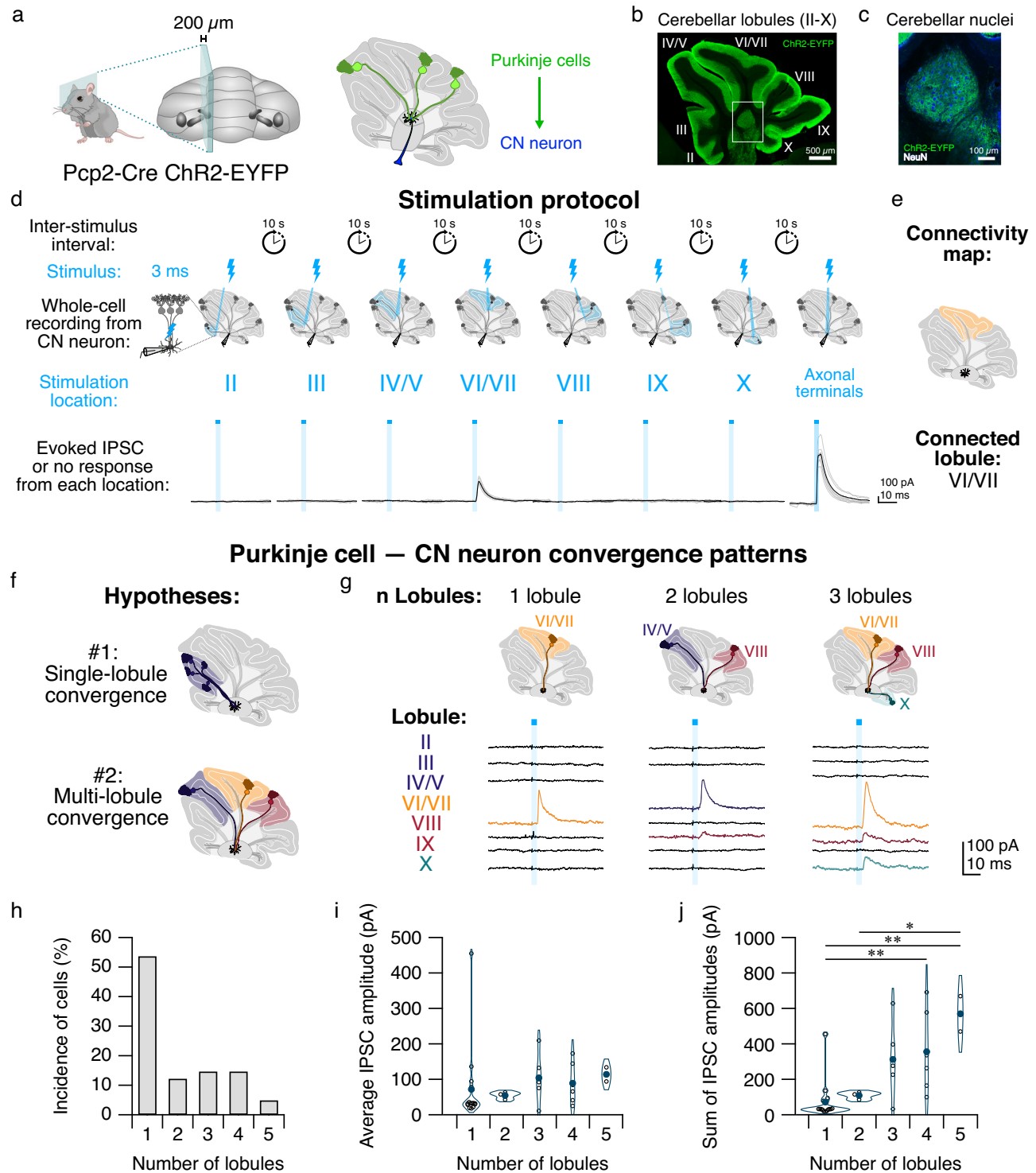

We next wondered whether the orientation of CN neuron dendrites was related to the location of their Purkinje cell inputs (Supplementary Fig. 7). We found that most cells had at least some dendrites oriented towards their Purkinje cell inputs, but that it was typically a small fraction of the dendritic tree. The amount of dendrite – zone overlap increased as the number of zones connected to a given CN neuron increased, likely due to the greater area covered by the input direction from multiple lobules, suggesting that the orientation of CN dendrites is not shaped by its input. We also did not observe a relationship between IPSC amplitudes or rise times and the degree of overlap, suggesting that dendrites are not preferentially oriented toward or away from larger lobule inputs, and that input from different lobules is not clustered on different regions of the dendritic trees.

The CN has previously been described to exhibit a topographic organization that corresponds with the post-synaptic targets of CN neurons[6,31,32]. We wondered whether CN neurons also showed topographic organization according to their input connectivity patterns. Using coordinates from the Allen Mouse Brain Atlas[33], we visualized our recorded cells in 3-D within the fastigial CN[34] (Fig. 4a, Supplementary Fig. 5). CN neurons did not appear to cluster by connectivity pattern (Fig. 4b, left). However, clustering of CN neurons based on the number of input zones was weakly apparent in our data (Fig. 4b, right).

**Fig. 1 | Cerebellar nuclear (CN) neurons receive both single- and multi-lobule input. a** Left, cartoon of a transgenic mouse showing location of 200 μm-thick acute slice. Right, Purkinje cells express Channelrhodopsin-2 (ChR2) fused with enhanced yellow fluorescent protein (EYFP), and project to CN neurons. **b** ChR2-EYFP (green) in Purkinje cells. **c** Expansion of box in **b** shows ChR2-EYFP expression is restricted to Purkinje cell puncta (green) in the CN, while CN neurons express NeuN (blue). **d** Stimulation protocol. Lightning bolts indicate a single 3 ms flash of 470 nm light focused on stimulation location. Middle, CN neurons were patch-clamped while Purkinje cell axons in each location (left, base of individual cerebellar lobules, which blue lobule highlighting stimulated lobule; right, axonal terminals) were stimulated. Bottom, the evoked IPSC was recorded following stimulation. Gray traces represent individual IPSCs, black traces indicate average. Blue rectangles indicate onset and duration of light pulse. Lobule average IPSC: 91.18 ± 10.53 pA; axonal terminals average IPSC: 340.82 ± 35.34 pA. **e** Connectivity

map representing Purkinje cell – CN connectivity from cell shown in **d**. **f** Representation of hypotheses of Purkinje cell convergence on CN neurons. Top, Purkinje cells originating from a single lobule converge on a CN neuron (hypothesis #1); bottom, Purkinje cells originating from multiple lobules converge on a CN neuron (hypothesis #2). **g** Connectivity maps (top) and average evoked IPSCs (bottom) from CN neurons receiving input from 1 (left), 2 (middle), or 3 (right) lobules. **h** Percentage of cells connected to n lobules. **i** Average IPSC amplitude per cell with n connected lobules. **j** Sum of IPSC amplitudes per cell with n connected lobules. IPSC amplitudes compared using two-tailed non-parametric multiple comparison Mann–Whitney $U$ test. 1 lobule vs. 4 lobules: $P = 0.009$; 1 lobule vs. 5 lobules: $P = 0.003$; 2 lobules vs. 5 lobules: $P = 0.03$. *$P < 0.05$, **$P < 0.01$. Unlabeled comparisons = not significantly different. $n = 41$ cells with functional connections out of 75 cells. Source data are provided as a Source Data file.

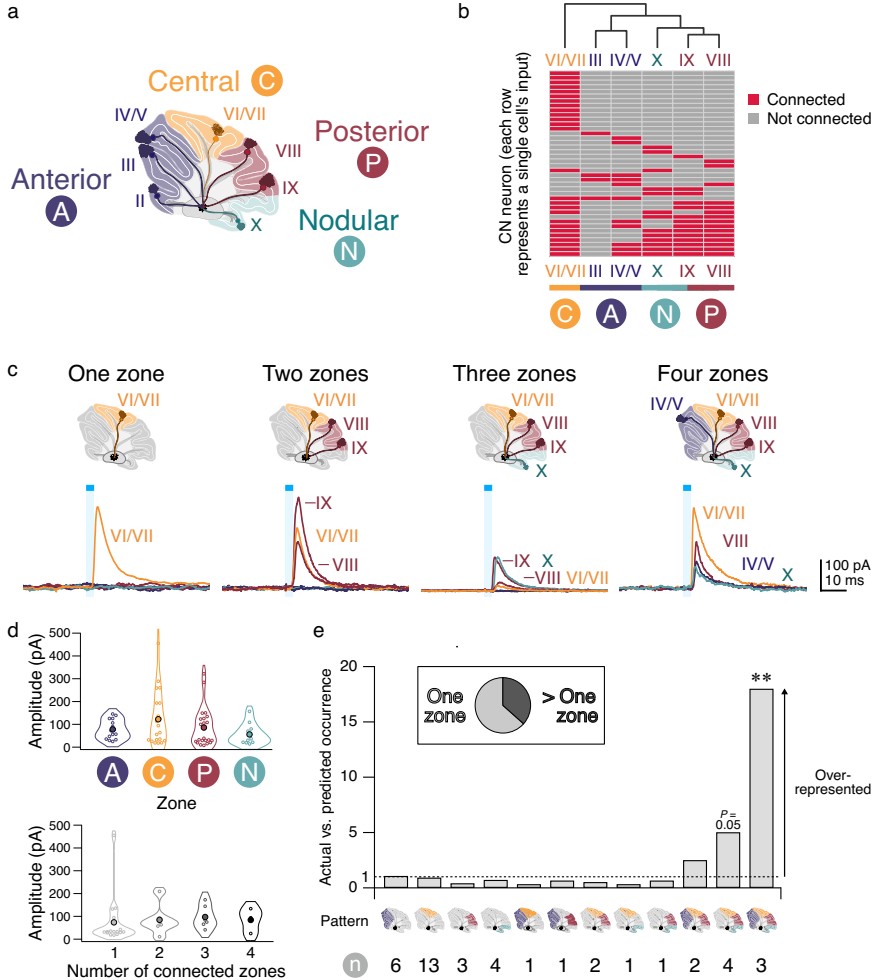

**Fig. 2 | Purkinje cell–CN neuron connectivity is not random. a** Cartoon of cerebellar lobules (roman numerals) and zones (circles). **b** Heatmap representing lobule inputs (columns) to individual CN neurons (rows) displaying connected lobules (red) and unconnected lobules (gray). Dendrogram of lobules created with unsupervised hierarchical clustering (top) using the Hamming distances between lobules' connectivity patterns with the unweighted average linkage reveals inputs cluster along functional zones (delineated by color, bottom). **c** Connectivity maps (top) and evoked IPSCs (bottom) from four CN neurons receiving input from one (left), two (middle left), three (middle right), and four (right) zones. Roman numerals beside traces indicate responsive lobule. **d** Top, average IPSC amplitude for each zone is not statistically different from other zones. Bottom, average IPSC

amplitude per cell based on number of connected zones. **e** Comparison of the actual vs. predicted occurrence of cells with each zone combination found in our dataset. Dotted line represents equal occurrence of predicted and actual combinations. One-tailed $P$ values represent the difference between the actual and predicted occurrence as determined using a binomial distribution test, with multiple hypothesis corrections performed using False Discovery Rate testing. Four zone over overrepresentation: $P = 0.009$. Inset shows proportion of CN neurons receiving input from one zone or multiple zones. IPSC amplitudes compared using non-parametric multiple comparison Mann–Whitney $U$ test. **$P < 0.01$. Unlabeled comparisons = not significantly different. $n = 41$ cells with functional connections out of 75 cells Source data are provided as a Source Data file.

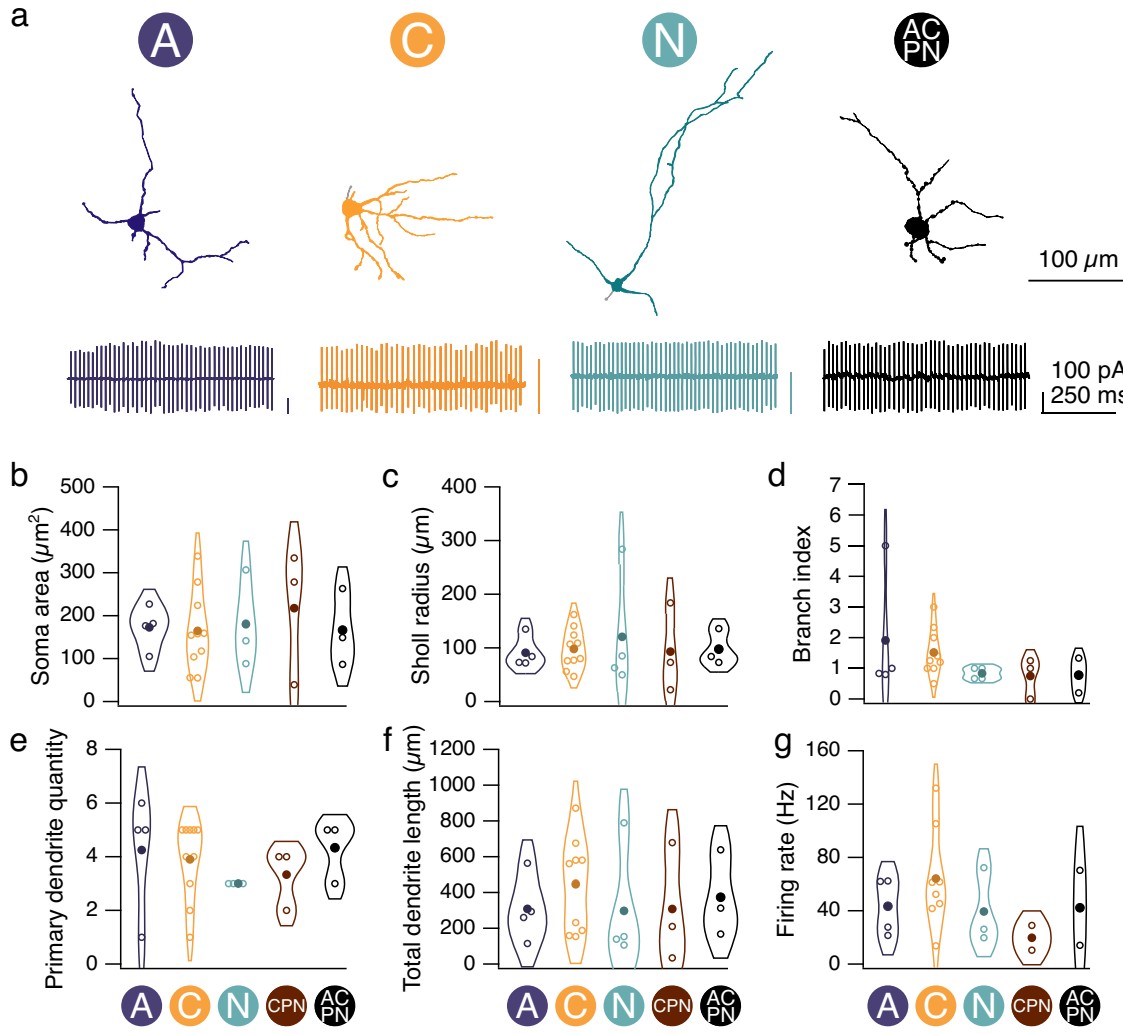

**Fig. 3 | CN neurons with different input connectivity patterns are morphologically similar. a** Top, reconstruction of CN neurons from two-photon stack. Circle denotes cell receives input from the anterior ("A") zone. Bottom, representative traces of spontaneous firing from CN neurons receiving input from the "A," "C," "N," or "ACPN," zones, respectively, obtained before breaking through into whole-cell configuration. For each trace, 100 pA y axis is indicated in corresponding color. **b**–**g** Soma area, Sholl radius, branch index, primary dendrite quantity, total dendrite length, and firing rate for cells with different connectivity patterns, groups with <3 reconstructed cells were excluded. Morphological properties compared across connectivity patterns using two-tailed non-parametric multiple comparison Mann–Whitney $U$ test. Unlabeled comparisons = not significantly different. $n = 24$ cells (**b**–**f**), $n = 18$ cells (**g**) out of 75 cells. Source data are provided as a Source Data file.

To determine whether topographic organization along any axes was observed (Fig. 4c, d), we looked at the fit of cells with different numbers of zonal inputs (e.g., 1 zone, 2 zones, 3 zones, 4 zones, Fig. 4e). We fit a line for each group of cells receiving input from $n$ zones and determined the $R^2$ of each. We then performed a bootstrap analysis to randomly sample an equivalent number of cells for each group from our total distribution 50,000 times. To determine how similarly the observed samples' fit matched our data, we assigned significance to any $R^2$ value from our data that was in the top 5% of distribution of bootstrapped $R^2$ values (Supplementary Fig. 8). A significant $R^2$ value was only observed once: for 4-zone cells along the rostroventral-caudodorsal axis (Fig. 4f), but not along the mediolateral axis (Fig. 4g, h). This suggests that 4-zone cells may show some topographic organization within the fastigial nucleus.

**Purkinje cell puncta from different zones terminate onto CN neurons in close proximity**

To complement our electrophysiological findings of multi-zone convergence on individual CN neurons, we used an anatomical

tracing approach. In fact, previous studies have identified anatomical convergence within longitudinal stripes from Purkinje cells originating from different lobules[35]. We performed stereotactic surgeries to inject up to 3 adeno-associated viruses (AAVs) expressing different fluorophores into different cerebellar zones of adult *Pcp2-Cre*-negative mice to label Purkinje cells and their axons. This enabled us to visualize the axons and terminals of Purkinje cells from different locations within the CN (Fig. 5a). We found that Purkinje cells from different zones (Fig. 5b, c) show gross differences in their projection patterns within the nucleus (Fig. 5d), as previously reported[31]. However, in many regions of the CN, we observed puncta from different zones in close proximity (Fig. 5e). In fact, ~20% of puncta from Purkinje cells originating in one zone were located within 17.5 µm of puncta from a different zone, which is the average CN soma diameter we measured (Fig. 5f, Supplementary Fig. 9) and >85% of puncta were located within 100.5 µm, the average length of a CN dendrite from our dataset (Fig. 5f, Supplementary Fig. 9). Using immunohistochemistry, we confirmed that many puncta expressed a marker for the vesicular GABA

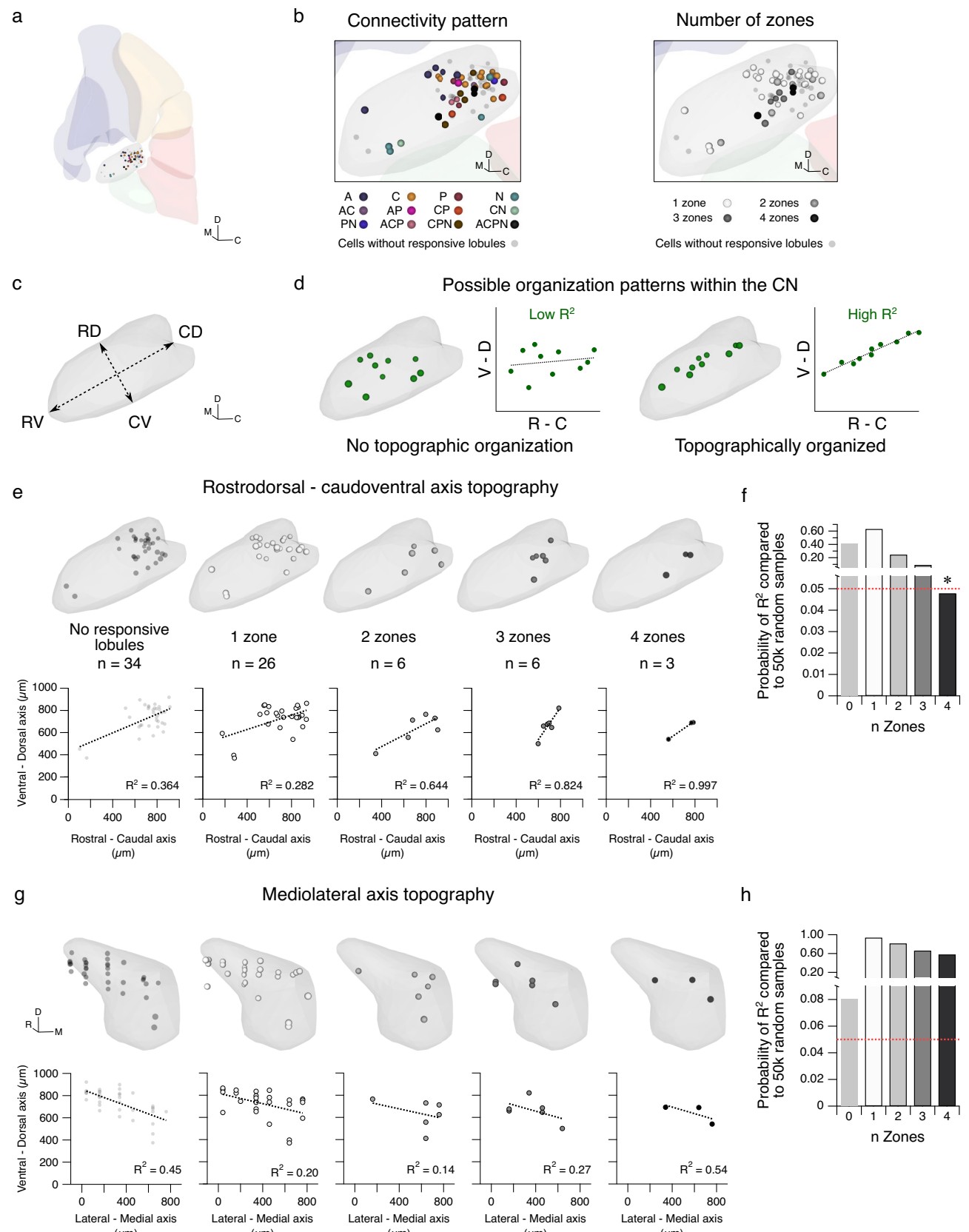

transporter (VGAT, Supplementary Fig. 10), confirming that many of the virally labeled puncta we visually identified are putative sites of Purkinje cell presynaptic terminals. Our findings suggest that Purkinje cells from different zones frequently terminate within

close enough proximity that they can synapse onto the same CN neuron, lending anatomical support to our optogenetic and electrophysiological findings of convergence across multiple functional zones (Fig. 2).

**Fig. 4 | CN neurons receiving input from all four zones are biased along the rostroventral-caudodorsal axis. a** 3-D representation of recorded cells using coordinates from the Allen Mouse Brain Atlas and rendered using Brainrender. **b** Left, expansion of inset in **a** depicting position of cells based on connectivity pattern. Right, cells in **b**, left categorized by number of input zones. **c** Rendering of fastigial CN depicting the rostroventral-caudodorsal (RV-CD) and rostrodorsal-caudoventral (RD-CV) axes. **d** Cartoons of two possible organization patterns within the CN, where one type of CN neurons receiving a pattern of Purkinje cell input do not demonstrate topographic organization (left, "Low $R^2$"), and another where CN neurons receiving a particular pattern of Purkinje cell input exhibit topographic organization biased along a particular axis of the nucleus (right, "High $R^2$"). **e** Top (from left to right), rendering of position in the sagittal plane of CN neurons that responded only to axonal terminal stimulation (No responsive

lobules), and CN neurons that received input from one zone, two zones, three zones, and four zones. Bottom, same as in **e**, top, but showing the plotted points for each CN neuron in the rostral-caudal and ventral-dorsal axes. Dotted lines and $R^2$ values in each graph represent the line of best fit for the neurons in each category. **f** Bootstrap analysis was used to assess the significance of observed $R^2$ fits in **e**, shuffling data for 50,000 iterations to produce a large distribution of $R^2$ values. Only the n Zones = 4 group has an $R^2$ value that fell within lowest 5% of bootstrap values (one-tailed, $P = 0.048$, not corrected for multiple comparisons). Red dotted line indicates $P = 0.05$. **g–h** Same as **e–f** but along the mediolateral and ventral-dorsal axes. *$P < 0.05$. Unlabeled comparisons = not significantly different. $n = 75$ cells. C: caudal, D: dorsal, M: medial, R: rostral, V: ventral. Source data are provided as a Source Data file.

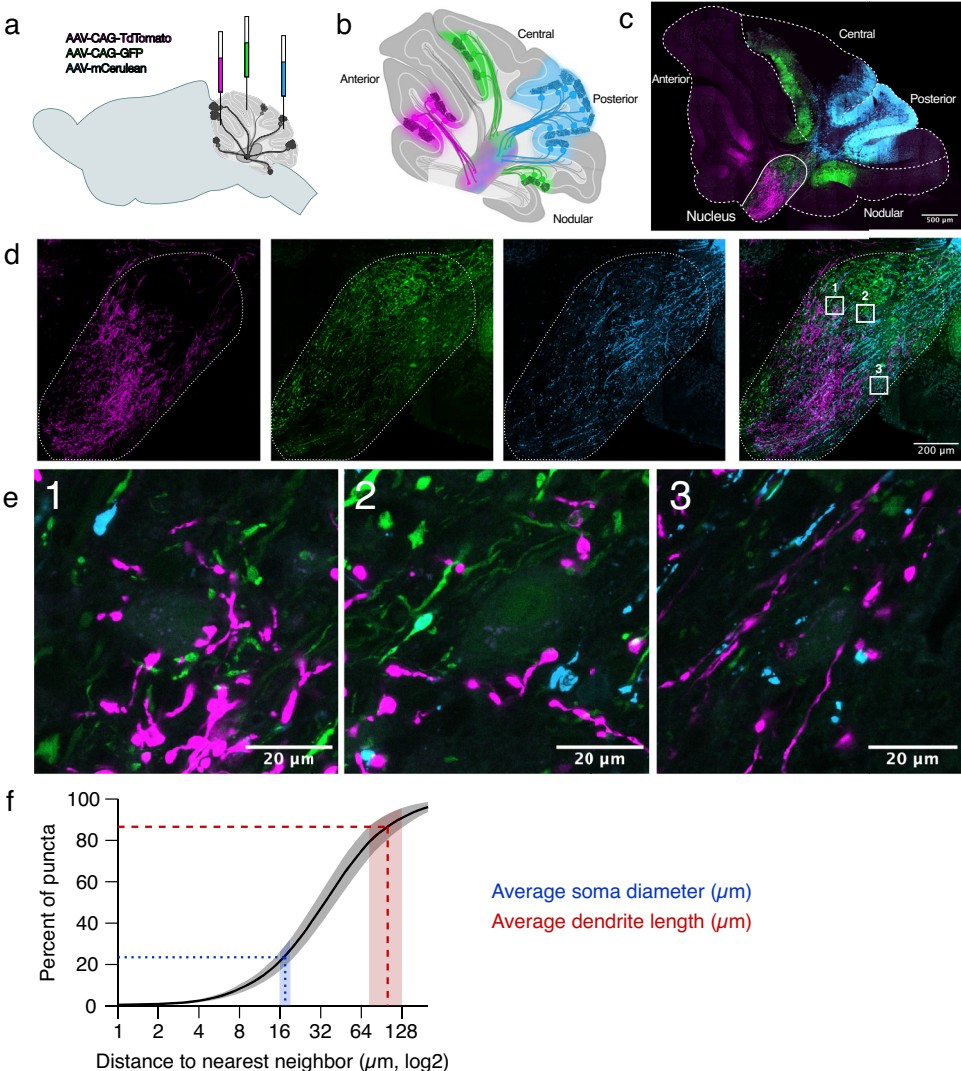

**Fig. 5 | Purkinje cell puncta from different zones terminate in close proximity in the CN. a** Cartoon representing intracranial injections of fluorescent viruses into different areas of the cerebellar vermis. **b** Cartoon representing Purkinje cells labeled with virus injections depicted in **c** representative image of cerebellum with viruses expressed in different zones at 20x magnification. **d** Expansion of nucleus in **c** showing that Purkinje cells from different lobules target different areas of the CN, 20x magnification. **e** Expansion of numbered boxes in **d** showing Purkinje cell

puncta from different zones in apposition to individual CN neurons, 63x magnification. **f** Cumulative probability of nearest neighbor distance of Purkinje cell puncta from different zones. Dashed lines: average soma diameter (blue, 17.5 microns) and dendrite length (red, 100.5 microns). Shaded rectangles: 95% confidence interval of nearest neighbor puncta across animals. $N = 6$ animals. Source data are provided as a Source Data file.

## Simultaneous activation of single-zone Purkinje cell inputs is sufficient to pause CN neurons

It has been suggested that Purkinje cell input synchrony is required to affect the output of CN neurons[5,36,37] (but see also ref. 38). Our

observation that multiple zones converge on single CN neurons raises the question of whether synchrony is required across functional zones to affect cerebellar output. To explore this, we recorded action potentials from CN neurons from a subset of spontaneously active

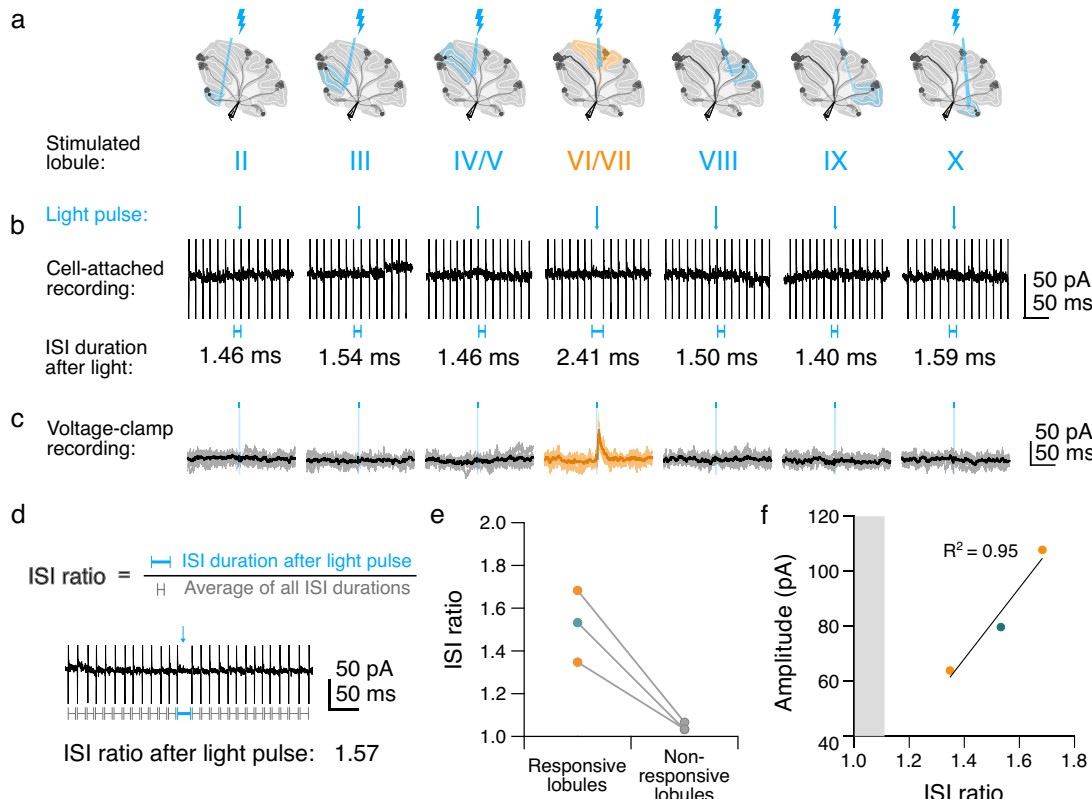

**Fig. 6 | Small inputs from a single lobule are sufficient to alter CN output.**
**a** Stimulation protocol. **b** Top, representative traces of spontaneous firing from a CN neuron during lobule stimulation before breaking into whole-cell configuration. Blue line with arrowhead indicates onset and duration of light pulse. Bottom, blue lines indicate duration of inter-spike interval (ISI) immediately following stimulation. **c** Voltage-clamp traces during stimulation from cell in **b** after breaking through to whole-cell configuration. **d** Top, ISI ratio calculated by dividing the ISI duration after stimulation with the average ISI duration per cell per trial. Bottom, for example trace from a cell with an ISI ratio of 1.57. **e** ISI ratios per cell for responsive lobules: 1.52 ± 0.17; non-responsive lobules: 1.04 ± 0.02. **f** ISI ratio vs. IPSC amplitude per cell, Pearson's $R^2$ correlation coefficient. Shaded area depicts 95% confidence interval of the ISI ratio of non-responsive lobules (≥1). Unlabeled comparisons = not significantly different. $n$ = 3 cells out of 75 cells. Source data are provided as a Source Data file.

cells (Fig. 6a, b) prior to breaking through into voltage-clamp configuration for IPSC recordings (Fig. 6c), and stimulated Purkinje cell axons as described above (Fig. 1d). This allowed us to measure both the impact of Purkinje cell stimulation on CN firing and the synaptic response within the same CN neuron. We measured the inter-spike intervals (ISI) of CN neurons immediately after the light pulse as a ratio of the average ISI duration per trial for each CN neuron that fired regularly (Fig. 6d, "ISI ratio"). We found that the ISI ratio immediately following stimulation of lobules that were connected to the CN neuron was typically higher than that for lobules that were not synaptically connected to the CN neuron (Fig. 6e). We also found that IPSC amplitude was positively correlated with the ISI ratio (Fig. 6f), where small inputs produced small pauses, and larger inputs produced larger pauses. While lobule stimulation likely activates more than one Purkinje cell axon, and thus may reflect multiple synchronous inputs, even small inputs in the range of ~60 pA (~50% of all recorded inputs from lobules ≤ 60 pA) were sufficient to pause CN firing, thereby influencing cerebellar output.

## Discussion

Our results reveal how CN neurons in the fastigial nucleus receive input from Purkinje cells that arise from multiple sagittal locations across the cerebellar vermis[19,20], and show that CN neurons are an overlooked locus of cerebellar integration. Using electrophysiology combined with focal optogenetic stimulation of Purkinje cell input, we mapped the spatial patterns of inputs onto CN neurons within the sagittal plane. We found that input from Purkinje cells tended to

cluster within the transverse zones of the cerebellar cortex, with roughly two-thirds of CN neurons receiving uni-zonal input. However, some CN neurons received input from multiple zones. Cells that received input from all four functional zones was observed more often than predicted by a random model, and these 4-zone cells appeared to be topographically located near the midline of the rostrodorsal−caudoventral axis in the CN. Corroborating these findings, we used stereotactic viral expression of multiple fluorescent proteins to provide morphological evidence that puncta from Purkinje cells in different functional zones are proximal to puncta from other zones within the CN. Lastly, we showed that small synaptic inputs could pause CN neurons, suggesting that widespread Purkinje cell synchrony may not be necessary to influence CN neuron output. Taken together, our results reveal that CN neurons are dynamic computational units that contribute more to cerebellar integration than was previously appreciated.

One of the limitations of our study is that connectivity is restricted to axons within the thickness of the sagittal slice preparation. While fastigial neurons receive most input from the vermis in a spatially restricted manner[19,20,31], this suggests that the connectivity patterns we report correspond to a lower limit, and that some multi-zonal cells may have been misclassified as uni-zonal due to missed input. Notably, the size of the dendritic trees did not differ across cells with different numbers of zonal inputs, suggesting that this was not a systematic artifact of slice preparation. Thus, despite our finding that 4-zone connectivity was over-represented, we may nonetheless underestimate the prevalence of these cells in the CN.

The cerebellum has been subdivided in several different manners. One example is that of cerebellar modules, where the mediolateral position of parasagittally-oriented stripes of Purkinje cells are connected to a specific area in the CN, which projects to a location in the inferior olive that projects back to the original Purkinje cells, resulting in a topographic cerebellar cortico-nucleo-olivary loop[11,12,39]. Cerebellar modules A, A1, and AX are most likely to correspond to those that we assayed in our experiments[16]. A second manner in which the cerebellum has been divided is into functional transverse zones. Our results demonstrate that several Purkinje cell convergence patterns exists within the cerebellar vermis even within narrow slices that likely encompass a single cerebellar module. Our results complement studies of cerebellar modules by identifying patterns of connectivity that are likely contained within individual cerebellar modules[40], although anatomical studies have predicted multizonal structured connectivity[7,9]. Indeed, stripes have previously been shown to span non-adjacent lobules with anatomical studies[35,41], which our functional data both expands and corroborates.

Our observation that input to CN neurons clusters based on functional transverse zones of the cerebellar cortex underscores the importance of zonal organization in the cerebellum. We observed that CN neurons receiving 4-zone inputs from Purkinje cells were observed more than would be expected from a uniformly random model. Multizonal convergence onto a CN neuron suggests a previously unknown site for multimodal integration and output. Such multimodal integration has been observed before at the input level of the cerebellum in the granule cells[42], and is also a feature of Purkinje cells that integrate large numbers of inputs in their extensive dendritic trees[35,43–45]. However, it has not been clear whether multimodal integration occurred in the CN.

4-zone CN neurons are over-represented in our dataset, yet still represent only a fraction of the total cells in the network. Since many axons are cut in the slice preparation, our data represent a lower, rather than an upper bound of their prevalence in the fastigial nucleus. How might such highly connected but sparse neurons function in the CN? There are several examples in the literature where highly interconnected but sparse neurons play a powerful role in brain, including hub neurons in the developing hippocampus which drive synchronous network activity[46], and trigger neurons in the neocortex[47]. It will be important in the future to explore the functional impact of multi-zonal CN neurons, which may function as "integrator neurons" in the cerebellar circuit.

Although the CN have been proposed to have evolved from duplication events of conserved cell types[20], it is possible that the patterns of connectivity we observe in the fastigial nucleus differ from those found in the interposed and dentate nuclei. Reasons for this may be the distinct three-dimensional features of each nucleus, as well as their different downstream targets[6,20,31,48]. Recently, Wang and colleagues identified a class of CN neurons in the fastigial nucleus that make excitatory synapses onto neurons in the inferior olive that is not found in the other CN[49]. This makes it important to determine whether the Purkinje cell input motifs that we observe in the fastigial nucleus are conserved in other CN.

A great deal of recent work has focused on the different classes of neurons of the CN, based on their projections, molecular expression, and physiological properties[6,20,30,50]. We wondered if different CN neuron classes might receive different input motifs but were unable to find evidence for this. However, our inability to detect morphological or physiological distinctions between classes of cells based on connectivity motifs does not mean that they do not exist. While properties such as firing rate have been described to differ for different classes of CN neurons, the distributions of firing rates for each category overlaps, meaning that large numbers are needed to differentiate between these cell types[30]. We would be unlikely to discern small differences in overlapping distributions between CN neuron classes because the number of neurons we have characterized for each connectivity motif is relatively low. However, the topographic enrichment of 4-zone input cells spread along the rostroventral-caudodorsal axis, where 4-zone input cells are tightly clustered in the orthogonal rostrodorsal-caudoventral axis, is likely to have implications for the output circuits of the CN[6,48], although they do not appear to represent any defined output cell population identified to date (e.g., ref. [6]).

Several different theories have set out to explain how the cerebellum processes sensorimotor information while producing complex behaviors. In the universal cerebellar transform hypothesis, the cerebellum uses a one-size-fits-all computation to process diverse input and contribute to motor and cognitive functions[51,52]. Alternatively, the multiple functionality transform hypothesis[53] posits that distinct tasks require distinct algorithms which may be mediated by factors such as Purkinje cell firing rate, molecular profile, or timing rules[53–55]. Since neither theory has incorporated a role for CN neurons in cerebellar processing[56], our findings suggest that theories of cerebellar function may need to be updated.

Both CN neurons and Purkinje cells fire spontaneous action potentials at high frequencies. One theory about the information transfer at these synapses is that if inputs from different Purkinje cells were activated asynchronously, individual inputs may not be sufficiently strong to impact CN output, while synchronous convergent Purkinje cells may produce stronger inputs more likely to influence CN output[5]. Our data show that relatively small inputs of ~60 pA can alter CN neuron output, which is smaller than the previously estimated unitary amplitude for Purkinje cell – CN neuron synapses[5]. Previous work showed that hyperpolarizing current injections of ~100 pA on CN neurons altered spontaneous firing rates, while single evoked inhibitory postsynaptic potentials (IPSPs) from Purkinje were less successful at affecting CN firing than trains of evoked IPSPs[57]. Although this appears in contrast to our findings where individual inputs alter the timing of CN neuron spiking, since Purkinje cells typically fire at high frequencies in vivo, it will be interesting to explore the impact of optically elicited IPSPs delivered at more realistic frequencies in freely spiking CN neurons.

Given that we commonly observed a variety of IPSC amplitudes on individual CN neurons that receive input from multiple zones (Fig. 2c, Supplementary Fig. 2), our data suggest that input from different zones may differentially impact CN neuron firing, but that small inputs may nonetheless have some effect on cerebellar output. However, while synchrony from convergent Purkinje cells that target a given CN neuron may not be necessary to affect CN neuron output, since larger inputs appear to affect output more strongly, synchronous convergent input may provide a more powerful means to alter CN neuron output. Furthermore, since CN neurons likely receive more inputs in vivo than in vitro, the impact of individual IPSPs is likely diminished in the behaving animal, making it hard to extrapolate from our findings. It will be important in future studies to determine whether synchronous multi-zonal Purkinje cell input is necessary to influence CN neuron output in vivo, as this will shed light on the role of CN information processing in the cerebellum.

## Methods

### Animals

We crossed mice hemizygous for Purkinje cell-specific Cre [strain B6.Cg-Tg(Pcp2-cre)3555Jdhu/J; stock number: 010536; PCP2-Cre] with mice with a loxP-flanked stop cassette upstream of Channel-rhodopsin-2(ChR2)/H134R fused with enhanced YFP [strain: B6;129S-Gt(ROSA)26Sor^tm32(CAG–COP4*H134R/EYFP)Hze/J; stock number 012569; Ai32] to produce hemizygous PCP2-Cre/Ai32 mice[21] that express ChR2 in Purkinje cells, or ChR2(H134R)-EYFP mice. Mouse strains were purchased from Jackson Laboratories. Mice were housed with a 12/12 h light/dark cycle (dark 7 pm to 7 am). Room temperature was maintained between 18–24 °C and 30–70% humidity. All animal procedures were approved

by the McGill Animal Care Committee, in accordance with guidelines established by the Canadian Council on Animal Care.

## Acute slice preparation

Acute sagittal slices were prepared[22] from 22 female and 25 male mice, aged postnatal day P21 to P32 (Supplementary Fig. 11). Mice were deeply anesthetized with isoflurane until unresponsive to a toe pinch and were then rapidly decapitated. Brains were removed and placed in partial sucrose replacement slicing solution (in mM: 50 NaCl, 2.5 KCl, 0.5 CaCl$_2$, 10 MgCl$_2$, 1.25 NaH$_2$PO$_4$, 25 NaHCO$_3$, 25 glucose, and 111 sucrose bubbled with 95% O2–5% CO$_2$, to maintain pH at 7.3; osmolality ~320 mOsm) heated to -37 °C[58], or in a subset of experiments, in ice-cold slicing solution. We used a Leica VT 1200 S vibrating blade microtome (Leica Microsystems, Wetzlar, Germany) to cut parasagittal slices of left cerebellar paravermis at a thickness of 200 μm in a chamber heated to 37 °C. Slices were incubated in artificial cerebrospinal fluid (ACSF; in mM: 125 NaCl, 2.5 KCl, 2 CaCl$_2$, 1 MgCl$_2$, 1.25 NaH$_2$PO$_4$, 25 NaHCO$_3$, and 25 glucose, bubbled with carbogen; osmolality ~320 mOsm) at 37 °C for 30–45 min in a darkened slice chamber[22], and subsequently moved to room temperature for up to 6 h. Chemicals were purchased from Sigma-Aldrich (Oakville, ON, Canada) except CaCl$_2$ and MgCl$_2$ from Fisher Scientific (Toronto, ON, Canada).

## Electrophysiology

Using acute sagittal slices from mice expressing ChR2 in Purkinje cells, we performed cell-attached and whole-cell patch clamp recordings from CN neurons from the fastigial (medial) cerebellar nucleus. To do this, we pulled borosilicate patch pipettes (2–7 MΩ) using a P-1000 puller (Sutter Instruments, Novato, CA, United States) and filled pipettes with an internal solution containing (in mM): 150 potassium gluconate, 3 KCl, 10 HEPES, 0.5 EGTA, 3 Mg-ATP, 0.5 GTP tris salt, 5 phosphocreatine-(di)tris, with 302 mOsm and pH 7.2 (adjusted with KOH). AlexaFluor 594 (50 mM, ThermoFisher, Burlington, ON, CA) was added to the internal solution to fill patched CN neurons to enable post-recording imaging and morphological reconstruction. Recordings were acquired with a Multiclamp 700B amplifier (Molecular Devices, Sunnyvale, CA, United States) on a SliceScope Pro 3000 microscope (Scientifica, Uckfield, United Kingdom) from CN neurons in slices maintained at a temperature of 34 °C ± 1 °C bathed with oxygenated ACSF.

We prepared and recorded from three consecutive 200 μm-thick slices containing the left fastigial nucleus, sampling from a total width of 600 μm per animal. To identify CN neurons for patching, slices were placed in the same orientation in the rig bath with the mediolateral position of each slice noted (Supplementary Fig. 12) to both ensure consistent CN neuron sampling as well as orienting the recording electrode out of the way of the photostimulation regions of interest. For each slice, we performed an identical line-by-line scanning process to visually identify CN neurons for patching, starting from the most ventral position of the fastigial nucleus and continuing to scan dorsally through the entire section of fastigial CN. For cell-attached recordings, we recorded from spontaneously-firing CN neurons before breaking through into whole-cell patch configuration. For voltage-clamp recordings, CN neurons were clamped at −60 mV and R$_{in}$ and resting membrane potential were monitored, and recordings were excluded if R$_{in}$ fluctuated more than 20%. Data acquisition was performed using custom Igor Pro acquisition software[22,59] (Wavemetrics, Portland, OR, United States).

## Optogenetics and spatial mapping

We used a Polygon400E patterned spatial illuminator with a 470 nm LED light source (Mightex, Toronto, ON, Canada) at an estimated focal plane power density of 100 mW/mm$^2$ through a 40X water-immersion objective (Olympus LUMPLFLN40XW, Tokyo, Japan) to optically stimulate Purkinje cell axons focally[22] using a blue rectangular (85 ×95 μm) light pulse of 3 ms with an inter-trial interval of 10 ms. To assess cell health, we stimulated the patched CN neuron to activate a large proportion of the synaptic inputs onto the cell directly (Purkinje cell "axonal terminals"). We next stimulated the base of each lobule sequentially (e.g., moving from lobule II to III to IV/V, etc. to lobule X, then over the axonal terminals, for ≥3 trials per location to identify functionally connected lobules. In a subset of CN neurons, we sampled connectivity by stimulating Purkinje cell axons at additional locations.

To assess the impact of synaptic input on CN neuron spontaneous firing, we performed cell-attached recordings in CN neurons and sequentially stimulated individual lobules as described above before breaking through into voltage-clamp mode. We excluded cells that broke through into whole-cell mode before stimulating each lobule at least once, trials in which spontaneous firing frequency was <20 Hz, and trials that were marked by bursting rather than tonic firing. After completing the stimulation protocol in cell-attached mode, we broke through and patch-clamped the CN neurons to compare voltage-clamp recorded IPSCs with their corresponding spontaneous firing data.

## Analysis of electrophysiological data

All electrophysiological data were analyzed using custom Igor Pro data analysis software[22,59]. For IPSCs, the rise time was measured as the time between 20–80% of the peak (standard deviation, SD)[22]. To assess the decay phases of synaptic currents, we used a double-exponential to fit the decay phases of IPSCs and calculated weighted decay τ time constants by assessing the contribution of each component of the exponential to the peak IPSC amplitude[23]. Proportion of failures was measured as the number of trials in which a responsive lobule failed to elicit an IPSC while the patch was still successfully in place as determined by the axonal terminal stimulation. For CN neurons that received input from more than one lobule or zone, we compared differences in amplitudes and rise times between individual IPSCs using a disparity index, or a measure of the coefficient of the variation of evoked IPSC amplitudes[24].

## Identifying connectivity motifs

To determine whether structure in connectivity patterns was observed, we performed two analyses. First, we created a binary dataset including each CN neuron's connectivity to lobules III through X, excluding lobule II since several lateral slices did not include lobule II, and excluding neurons from which we did not identify a connected lobule ($n = 34$ cells). We then performed unsupervised hierarchical clustering of the cells with identified functional connections ($n = 41$ cells) to evaluate potential patterns of connections between Purkinje cells in individual lobules and a given CN neuron, where Purkinje cells in lobules that belong to a shared clade (e.g., lobules IX and VIII) are more likely to be associated with the same CN neuron than those in lobules that are not located within the same clade (e.g., lobules VI/VII and X). Leaf and clade length denote level of similarity between leaves, where leaves and clades of shorter lengths denote lobules that have a higher likelihood of being functionally connected to the same individual CN neurons. We used Seaborn to create a clustergram of the binary connections between cells (Fig. 2b, rows) and lobules (Fig. 2b, columns) based on the Hamming distances between lobules with an unweighted average linkage. Input from lobules II, III, and/or IV/V were categorized as the anterior zone; lobule VI/VII as the central zone; lobule VIII, IX (without lobule X) as the posterior zone; input from lobule IX and X, or only lobule X, as the nodular zone.

To identify whether a given pattern of input to a CN neuron constituted a motif, or a statistically overrepresented connectivity pattern in our dataset, we tested the following hypothesis. From adequately sampled CN neurons, we assumed input from each of the four cerebellar zones to occur independently of input from another zone.

We calculated the probability of input from each zone within the dataset independent of each other probability. To evaluate multi-zone connectivity patterns, we multiplied the probability of each zone occurring in our dataset to the probability of the other combination(s), multiplied by the probably of non-connection to excluded zones. For example, a Central-Posterior input probability = the probability of input from the central zone (26 cells/75 cells tested) * the probability of input from the posterior zone (16 cells/75 cells tested) * by the probability of not-A anterior zone (1–13/75 cells) * the probability of not-N nodular zone (1–13/75 cells) = (0.347 * 0.213 * 0.827 * 0.827). We then compared the actual probability of the combinations occurring in our dataset to the predicted using a binomial probability test, where combinations that occurred significantly more or less often than predicted were described as motifs.

Unless otherwise indicated, analyses performed in Figs. 1–4 and 6 were performed on the same sample of $n = 41$ cells with lobule connectivity identified, out of 75 CN neurons.

## Intracranial viral surgeries

To label Purkinje cell axonal projections to CN neurons, we first injected *Pcp2*-Cre negative mice ($N = 6$ female mice, ages 6–14 months) intraperitoneally with a 25% solution of mannitol in PBS (7.5 g/Kg; Sigma-Aldrich, Oakville, ON, Canada) in order to enhance virus distribution[60,61]. Mice were then anesthetized using isoflurane (2.5% mixture in O2) before their fur was shaved and they were transferred to a stereotactic frame (Stoelting Co). Carprofen was given subcutaneously, and a lidocaine/bupivacaine mixture was applied locally for analgesia. We performed three injections on each mouse from posterior to anterior. Under stereotactic guidance, a small craniotomy was drilled (<1 mm) at the first location using a dental drill. A capillary filled with mineral oil (Sigma-Aldrich, Oakville, ON, Canada) attached to a nanoinjector (WPI, Sarasota, Florida, USA) was filled with 150–250 nL of virus, injected into the posterior cerebellar vermis (lobule VIII) and allowed to diffuse for five minutes. Once retracted, the mineral oil-filled capillary was replaced and realigned with Bregma. The second and third injections were performed in the central (lobule VI/VII) and anterior vermis (lobule IV/V) as described. The injection coordinates for surgeries were as follows along the midline ($X = 0$) from Bregma ($Y, Z$): Posterior vermis: (−7.25, −4.0 or 7.25, −4.25); Central vermis (−6.9, −2.5 or −6.9, −1.8 or −6.65, −1.8); Anterior vermis (−5.65, −3.0 or −5.7, −2.0 or −5.85, −2.5). We used the following viruses to label Purkinje cell axons arising from different cerebellar zones: pENN.AAV.CB7.CI.mCerulean.WPRE.RBG (titer ≥ $5 \times 10^{12}$ vg/mL), pAAV-CAG-tdTomato (codon diversified; titer ≥ $1 \times 10^{13}$ vg/mL) and pAAV-CAG-GFP (titer ≥ $7 \times 10^{12}$ vg/mL) viral preps were purchased from Addgene (Addegene viral prep #: 105557-AAV9, 59462-AAV8 and 37825-AAV8). After surgeries, mice were allowed to recover for 2–3 weeks before being euthanized and perfused as described below.

## Immunohistochemistry and viral labeling

We performed immunohistochemistry to demonstrate that CN neurons do not express ChR2 by perfusing mice and labeling for NeuN and anti-green fluorescent protein (GFP). For both immunohistochemistry and viral labeling experiments, we prepared slices from animals by deeply anesthetizing mice with 2,2,2-tribromoethanol (avertin, 0.02 mL/10 g) via intraperitoneal injection, followed by intracardiac perfusion[62]. We performed an initial flush using PBS and 5.6 µg/ml heparin salt, followed by perfusion with 40 ml of 4% PFA in phosphate buffer (PB, pH 7.4). We stored brains in 4% PFA for two days at 4 °C on a shaker at 70 RPM, followed by storage in PBS with 0.5% sodium azide prior to slicing. We collected sagittal slices (100 µm thickness) using a Vibratome 3000 sectioning system (Concord, ON, Canada). To label slices with NeuN and anti-GFP, we incubated slices for thirty minutes in blocking solution (1x PBS, 0.1 M, pH 7.4; 0.4% Triton X; 5% bovine serum albumin (BSA); and 0.05% sodium azide) which was followed by

a 3-day incubation with the primary antibodies in blocking solution. During primary antibody incubation, CN neurons were labeled with guinea pig anti-NeuN primary antibody (1:500, abN90, Millipore, Burlington, MA, United States) and rabbit anti-GFP primary antibody (1:500, ab3080, Millipore, Burlington, MA, United States). Slices were subsequently labeled with DyLight 405 anti-guinea pig secondary antibody (1:500, 106475003, Jackson Immunoresearch) and goat Alexa 594 anti-rabbit secondary antibody (1:1000, A11037, Life Technologies) during a 90-min incubation in blocking solution. We then rinsed and immediately mounted slices onto glass slides with Prolong Gold Antifade (ThermoFisher Scientific, Waltham, MA, United States) under low light, then stored slides in the dark at 4 °C.

## Image acquisition and analysis

Immediately after electrophysiology experiments, we performed live two-photon imaging of AlexaFluor 594-filled CN neurons on a custom-built two-photon microscope (Scientifica) imaged with a Ti:Sapphire laser (MaiTai; Spectra Physics, Santa Clara. CA, USA) tuned to 890 nm[63]. Images were acquired at a resolution of 512 by 512 pixels using ScanImage 3.7 running in MatLab 2011B (Mathworks, Natick, MA, USA). We later re-imaged slices containing filled CN neurons using a LSM800 laser scanning confocal microscope at 10x and 20x using Zeiss Zen software (Zeiss, Oberkochen, Germany) following immunohistochemistry staining and fixation in 4% paraformaldehyde (PFA). Slices labeling Purkinje cell puncta (anti-GFP) onto CN neurons (anti-NeuN) were imaged using the same confocal microscope using a 488 nm and 405 nm diode laser, respectively, to distinguish between the rabbit anti-GFP anti-rabbit Alexa594 labeling (Purkinje cell puncta, pseudocolored in green), and guinea pig anti-NeuN anti-guinea pig DyLite 405 labeling (CN neurons, pseudocolored in blue).

After imaging prepared slices, we used Fiji (ImageJ, NIH)[64] to analyze filled, stained CN neurons from electrophysiology and immunohistochemistry experiments. We confirmed the identity of Alexa-Fluor 594-filled CN neurons by comparing them with the initial two-photon images acquired following live electrophysiology experiments and identified the dendritic and lobule orientation in slices with filled cells to evaluate the relationship between CN morphology and input patterns. We used Neurolucida software (MBF Biosciences, Williston, VT, USA) to manually perform morphological reconstructions of CN neurons from the two-photon imaging session following patch clamp experiments. We traced cells in 3-D to obtain the cell soma contours, dendrite projections, and axon within the image stack. We imported tracings into Neurolucida Explorer to extract morphological data, including soma area, soma diameter, number, and length of dendrites. Branch index was calculated by dividing the number of dendritic crossings 25 µm from the cell soma by the number of primary dendrites[30]. Polar plots were made using Igor Pro to evaluate the overlap between dendritic orientation and Purkinje cell input. Dendritic polar plots were normalized to the length of the longest dendrite.

To evaluate the proximity of virally labeled Purkinje cell puncta originating from different cerebellar zones following intracranial stereotactic surgeries, we imaged mounted 100 µm-thick sections using the LSM-800 confocal microscope at 10x, 20x, and 63x oil immersion and acquired tiled stacks using a 405 nm, 561 nm, and 488 nm laser to distinguish between the mCerulean, TdTomato, and GFP expression. Images were pseudocolored in blue, magenta, and green, respectively, and brightness and contrast values were enhanced for representative images. Raw images were thresholded in ImageJ and used to identify the X and Y coordinates of each puncta in individual channels. Using a custom Python script, we identified the nearest neighbor puncta across different zonal inputs with different imaging channels to identify the Euclidean distance of the nearest puncta across zones. To confirm that there was little to no colocalization of multiple viruses in the same Purkinje cells, we performed thresholding of Purkinje cell

puncta for each channel in ImageJ and acquired the positions of puncta in the X-Y plane using ImageJ's plot profile analysis function and used a custom Python script to identify the percentage of colocalized pixels per slice in each stack which determined there was virtually no colocalization between channels (Table S1).

## Cell topography within fastigial nucleus

The position of each recorded CN neuron (Fig. 4) was recorded manually during each experiment and paired with an image of the position of each patch pipette during the recording. We mapped these positions for each slice along the mediolateral axis using the coordinate system from the Allen Mouse Brain Atlas[33], and for cells that were successfully filled with Alexa594 during each experiment, we confirmed these positions after confocal imaging of each slice. We then used Brainrender[34] to visualize these cells in 3-D within the nucleus. Next, we represented plotted CN neurons based on their number of input zones (0 to 4 zones) along the Allen Mouse Brain Atlas coordinates and produced graphs along the rostro-caudal and ventral-dorsal axes. We performed a linear regression for each category ($n = 0, 1, 2, 3,$ or 4 zones) and identified the $R^2$ value for each group. To evaluate whether the strength of the best fit line for each category was significant, we performed a bootstrap analysis where we randomly sampled n cells (equivalent to the number of cells in our dataset that received input from n zones) from each category 50,000 times and determined whether the $R^2$ value from our data occurred within the top 5% of $R^2$ values from the bootstrapped samples (one-tailed test). We plotted the percentile of the $R^2$ values from our data compared to the bootstrapped samples as the "probability of $R^2$ compared to 50,000 random samples" (Fig. 4f, g).

## Statistics

We used JMP software (SAS, Carey, NC, United States), to perform Student's t-tests on datasets with normal distributions, and Igor Pro software for non-parametric Mann–Whitney *U* tests for datasets with non-normal distributions, with the level of significance (α) set at $P < 0.05$. Unless otherwise indicated, all statistical tests performed were two-tailed tests. To evaluate connectivity motifs, we used a binomial distribution calculator followed by multiple hypothesis corrections using the Benjamini–Hochberg method. Data are reported as mean ± SEM unless otherwise indicated. For all data, $n$ = number of neurons and $N$ = number of mice.

## Reporting summary

Further information on research design is available in the Nature Portfolio Reporting Summary linked to this article.

# Data availability

All relevant data are available from the corresponding author upon request. The data generated in this study have been deposited in the Zenodo database under accession code 11090248. All data are available under restricted access, access can be obtained by contacting the corresponding author and explaining the intended use of the data. The raw data are protected and are not available due to data privacy laws. The data generated in this study are provided in the Source Data file. The datasets analyzed in this study are available on Github under accession code StructuredConnectivity. Source data are provided with this paper.

# Code availability

Custom code used to analyze the findings of this study that has not yet been described in published literature is available on GitHub: (https://github.com/kgruver/StructuredConnectivity).

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

## Acknowledgements

We thank D. Bowie, J. Lefebvre, and T. Ohyama for helpful advice. We thank C. Chou for training and technical support. We thank C. Vaaga, N. Al-Sharif, and J.W. Vogel for advice about data analysis. We thank all current and former members of the Watt laboratory for advice and support, especially S. Jayabal, D. Lang-Ouellette, and A. Cook. We thank the staff at the Advanced BioImaging Facility (ABIF) at McGill University for their technical support with confocal imaging. We thank A. Hargreaves and students from BIOL 610 at McGill for helpful comments and support in drafting an early version of this manuscript. We thank T. Koch and the other team members from the Comparative Medicine and Animal Resources Center (CMARC) for their technical support. This work was supported by the following: NSERC Discovery grants (RPGIN-2016005118 and RGPIN-2022-03439) and CIHR project grant (PJT-153150) (A.J.W.), a Fonds de recherche du Québec – Nature et technologies (FRQNT) Doctoral Scholarship, a McGill Integrated Program in Neuroscience Graduate Excellence Award, and a Graduate Student Fellowship from the Canada First Research Excellence Fund Healthy Brains for Healthy Lives initiative (K.M.G.), and a Fonds de recherche du Québec – Santé (FRQS) Doctoral Scholarship (E.F.).

## Author contributions

K.M.G. designed and ran experiments for all figures except Fig. 5, analyzed data for all figures, and wrote the manuscript.; J.W.Y.J. analyzed data for Figs. 3–4, S4–5, and S7; E.F. designed and ran experiments for Fig. 5; S.S. analyzed data for Fig. 2; P.J.S. wrote custom Igor software packages for electrophysiology data acquisition and analysis, and analyzed data for Figs. 2 and 4. A.J.W. conceived of the project, designed experiments, analyzed data, supervised the project, and wrote the manuscript.

## Competing interests

The authors declare no competing interests.
