## [Peer Review File · Nature Communications]

Structured connectivity in the output of the cerebellar cortexReviewers' Comments:

Reviewer #1:

Remarks to the Author:

This manuscript from Alanna Watt and her group shows that neurons in the cerebellar nuclei (CN neurons) may receive input from Purkinje cells located in different cerebellar lobules. This seems to be the case in roughly 50 percent of target neurons, while the remaining neurons get their input from Purkinje cells located in one lobule only. The study provides a connectivity map that is based on optogenetic activation of Purkinje cell axons and electrophysiological recordings from CN neurons in slice. Finally, the authors investigate how much current is needed to affect spike firing in CN neurons and come up with a low number of about 60pA. This work is very interesting and has the potential to have a very high impact on the field. As always for papers from this lab, the work is excellent in that it is well documented, beautifully illustrated and just a joy to read. There are two main comments that I have about the core claims that are made here:

1) The key message of the manuscript is that Purkinje cells from different cerebellar lobules can converge on the same target neurons in the cerebellar nuclei. If true, this finding will have substantial consequences on how we see computations at the output stage of the cerebellum arising. In that light, it is somewhat surprising that the authors do not offer controls that would exclude the possibility that blue light activation accidentally co-activates axons arising from other lobules. I did not find any control or comment about this, other than a reference to a Frontiers paper, but when looking that up, it also does not address the issue. However likely or unlikely this accidental co-activation is about to happen, a control needs to be provided. There might be different ways to do this. I could imagine one of these two working, but others might work as well: a) if optogenetic axonal activation elicits antidromic spikes that can be measured in Purkinje cell somata, the authors could test whether Purkinje cells in adjacent lobules ever get activated by axonal stimulation in the neighbor lobule. If so, this would mean that by axonal stimulation, the authors were not truly able to separate Purkinje cell output by lobule. And b) if such antidromic spikes are not seen and thus cannot be used as a tool here, the authors could pick up on an observation from their Frontiers paper. A threshold duration of the blue light pulse is needed for activation. If there is unintended cross-talk of optogenetic activation, applying a subthreshold duration light pulse to axon bundle A and another to B (sequentially) would still lead to CN neuron activation, because of the cross-talk between A and B. There might be other tests to probe for unintended cross-activation, but a control must be provided.

2) Fig 6: the conclusion from the data presented that an unexpectedly small amount of current is sufficient to affect spike firing is a bit puzzling. First, earlier recordings from DCN neurons have shown that starting with about 100pA currents, these neurons fire rebound spikes (Aizenman / Linden). Not the same experiment, but it also shows that currents in the same order of magnitude alter spike firing in these neurons. That means this observation – at the level of slice recordings – is neither surprising nor new. Second, it is not clear to me how informative this finding is. An (artificial) light pulse or current injection of 60pA has a different effect than, for example, a Purkinje cell complex spike, for a multitude of reasons, including the very different kinetics. Finally, the question is whether this value would be different when recorded in vivo. For these reasons, the authors should provide more appropriate context when discussing their finding, and they should be open to discuss the technical limitations here.

Reviewer #2:

Remarks to the Author:

In this MS, the authors recorded neurons in the medial cerebellar nucleus in brain slices while stimulating Purkinje cell axons at the bottom of individual cerebellar lobules using optogenetic. They found that single cerebellar neuron often receives Purkinje cell inputs from different lobules belonging to different antero-posterior zones. Connectivity pattern were found non-random, but no clear relationships between connectivity patterns and functional or morphological features were observed.

Finally, the activation of Purkinje cells in one lobule is enough to induce a pause in nuclear neurons. Overall, this MS is interesting as it addresses functional cortico-nuclear connectivity in brain slices using optogenetic stimulation of Purkinje cells in individual lobule. The authors should be commended for this original way to assess cerebellar physiology. However, technical limitations associated to the poor contextualization of the data in the literature of cerebellar modules limit the scope of the MS

Main points :

-In this MS, the authors emphasized the antero-posterior zonation of the cerebellum, with gross anatomical distinction such as anterior, central, posterior and nodular. These zones have certainly a developmental and physiological relevance as they were associated with specific task and behaviors both in rodents and humans (from being motor in the anterior to oculomotor in the central and non motor in the posterior parts). However, since decades now, the functional zones of the cerebellum are defined longitudinally (i.e. in the medio-lateral axis) as a combination of olivo-cortical and corticonuclear projections that define cerebellar modules and more precisely microzones. Cerebellar modules encompass several lobules, which are not necessarily contiguous (e.g. lobule III-IV with lobule VIII, Pijpers et al J Neurosci 2008, Ruigrok et al EJM 2008). Assumption from line 86: "we hypothesized that Purkinje cells in lobules belonging to the same functional zone were more likely to converge on a CN neuron than Purkinje cells in lobules from across zones" is not fully relevant based on the literature. Jan Voogd, Tom Ruigrok, Carl-Fredrik Ekerot, Henrik Jorntell, Richard Apps, Richard Hawkes, Izumi Sugihara and many other laboratories have spent years to study and refine the notion of anatomo-functional units in the cerebellum. In that respect, the multi-lobular Purkinje cell projection to cerebellar nuclei has been shown already several times (Voogd and Ruigrok, 2004, their Fig 4. Also shown by the group of Sugihara) using anatomical tracers. Actually, based on the maps from Sugihara's lab (J Neurosci 2004 and 2007) or from Ruigrok's lab (Ruigrok et al, EJM 2008), one could make precise predictions on PC convergence in cerebellar nuclei. Since the authors rather confirm anatomical predictions, at least an exhaustive comparison with the literature is necessary.

-As an extension of the first point, the vermis is composed of several sagittal zones: A1, Ax, and B from midline to lateral areas. Each of these zones have specific nuclear projections. For example, the lateral parts of A1 and B zones do not project to the medial nucleus but to the lateral vestibular nucleus (LVN) outside the cerebellum. Ax has also specific projections. Therefore, in the sagittal slices used in this MS, it is important to identify their medio-lateral position. A1+B zones are > 600 μ m wide, which means at least 3 slices from this study. Mediolateral information would help making sense of the distribution observed.

specific points:

-If I understand correctly, there are 3/71 (3/41 connected ones) neurons receiving Purkinje cell inputs from 4 zones, which means around 5%. I am afraid that they cannot be considered as an important group as is.

-I don't fully understand the rationale of the hierarchical clustering of Fig2b. It seems that lobule VI/VII is distant from others essentially because half of the neurons in this group were only connected to this lobule. Also, if we exclude the first half of the plot, in which neurons are connected to a single lobule, in the other half, they are connected to different antero-posterior zones. Typically, lobule VI/VII and at least one other. Hence, conclusion stated line 91 does not seem to be supported by the figure. Maybe machine learning algorithms (non-supervised or supervised) using several parameters as features (identity and number of connected lobules, position in the CN, firing rate....) could help isolating specific clusters of neurons even if simple correlations (Figure 3) could not.

-Most of the neurons seem located in the caudo-dorsal part of the medial nucleus, does it explain why half of them receive inputs from lobule the central zone?

-Why recordings were made at -60 mV in potassium gluconate internal solution, which is then very close of the chloride reversal potential if junction potential is added (around -10 mV more)? Could connected sites be missed?

-In previous studies, it was shown that the decay time constant of the GABAergic current from Purkinje cells to principal cells in the nuclei was ultra fast (Person and Raman, Nature 2011), while much slower at the Purkinje cell to nucleo-olivary neurons (Najac and Raman). Could the author evaluate the dataset in function of the decay time constant? Also, excitatory vs inhibitory neurons can be distinguished by their firing rates and spike shape (Ozcan et al, J Neurosci 2020 and Uusisaari et al

J Neurophysiol 2007), is it possible to sort them by this feature?

Reviewer #3:

Remarks to the Author:

Gruver et al have submitted a manuscript entitled, "Structured connectivity in the output of the cerebellar cortex." This paper seeks to evaluate a long held question of whether cerebellar nuclei neurons integrate information from multiple regions of the cerebellum. The authors perform a rigorous set of experiments – using slice physiology to examine Purkinje cell inputs into the cerebellar nucleus. It is known that signals from multiple Purkinje cells must be integrated at the level of cerebellar nuclear neurons because the number of Purkinje cells is >50x times greater than that of cerebellar nuclear neurons. However, the anatomical organization of this Purkinje cell convergence has not been well described – with numerous outstanding possibilities including but not limited to whether nearby PCs make convergent inputs, whether PCs belonging to the same module [Ito, 1984] make convergent inputs, or whether other forms of integration exist including connections from diverse regions across the cerebellum. These fundamental questions could shed light on cerebellar function and how cerebellar nuclei process information from the cerebellar cortex.

To get at this question, the authors combine slice physiology with focal optogenetic stimulation in vermal slices. They demonstrate that spatially segregated Purkinje neurons in different cerebellar lobules can make convergent inputs into a single fastigial/medial nuclear neuron and raises the possibility that previous findings across multiple species pointing to duplicated or triplicated functional topography within the cerebellum might be mediated by convergent Purkinje to cerebellar output neurons. These findings are important and are performed for the most part in rigorous fashion.

Several points, however, should be raised. Although these findings are important and could reveal fundamental insight into cerebellar function, the paper remains limited to this finding. The authors show the existence of convergence but do not provide functional insights into the function of these findings. Many potential questions could be addressed including but definitely not limited to: is the convergence necessary? Is the convergence redundant, what is the functional significance of this convergence, how generalizable is this finding outside of the limited subset of cerebellar nuclear neurons evaluated. Although adding new experiments of comprehensive in vivo functional analysis and/or anatomical analysis could resolve the current limitations, the amount of work needed for any of these directions would be considerably large to perform in the timeframe for the revision.

Specific comments.

1. The authors conclude that there are distinct fastigial neurons that (1) receive inputs from a single zone and (2) receive inputs from multiple zones. However, because of the limitation of the slice thickness being 200 um, the true existence of (1) as well as subdivisions within (2), remain in question. To their credit, the authors address this limitation in the discussion and note that this may limit the conclusions that can be drawn from this study despite still making the very conclusion that is made. The organization of the current manuscript, then, could be clarified to minimize confusion and to prevent any overstatement of manuscript findings.

2. It is also important to note the cerebellar nuclear neurons that were recorded are a very small group of neurons localized to a very specific caudal region within the medial nuclei (figure 4, supp figure 6). This fact makes it difficult to address whether the findings made are specific to this population. Also, this region is known to be comprised of morphologically similar neurons (Golgi staining analysis: Beitz and Chan-Palay, 1979, Neuroscience), which could contribute to the morphologic similarities noted by the authors. Importantly, the authors also suggest that CN neurons receiving diverse input are biased to caudal fastigial regions. However, with the specific sampling done, it's also quite possible that sampling bias could contribute to these findings. In fact, it is known that the rostral fastigial also receive inputs as a population from multiple vermal lobules (Sugihara et

al 2009); not having these regions tested could thus impact the strength of conclusions that can be made. In conjunction with the concerns raised above, it raises concern about generalizability and/or the extent of conclusions that can be made.

3. The authors note that their findings suggest that “widespread Purkinje cell synchrony may not be necessary to influence cerebellar output”, by showing that even small input stimulation can induce pauses in firing. However, the authors to my knowledge do not show what that small input/stimulation is actually resulting in in terms of Purkinje neuronal stimulation. Synchrony as determined by the referenced study by Person and Raman 2011 show that even a small amount of synchrony – even 2 afferents – can be sufficient. Thus, it is not clear whether these data are sufficient to support this claim.

4. As alluded to above, these data are not put into a functional or anatomical context either of which would dramatically affect the impact of these data. The AZ, CZ, PZ, and NZ are the classifications originally developed to indicate that these multiple zones harbor different Purkinje subtypes (Ozol et al., 1999, *J Comp Neurol*; Sillitoe and Joyner, 2007, *Annu Rev Cell Dev Biol*). At a circuit level, it is hypothesized that Purkinje neurons that make convergent input to the CN are located within the same cerebellar stripes (Sugihara et al., 2009, *J Comp Neurol*; Fujita and Sugihara, 2013, *Front Neural Circuits*). How the authors’ findings map on to this modular organization within the cerebellum is not clear from the manuscript.

5. The anatomical demonstration of convergence in Figure 5 could be more convincing. Since the dendrites of nuclear cells are densely packed, ~20 um proximity, as the authors show, the data do not necessarily indicate input to the cells of interest. Indeed, it’s potentially feasible that signals are not necessarily of terminals but of axonal segments that may be passing through the plane.

REVIEWER COMMENTS

We thank all the reviewers for their comments and feedback. We have addressed the reviewers' concerns below and in the manuscript. In summary, we have conducted new experiments, we have rewritten the manuscript, and we have included new figure panels and four new Supplementary figures in response to their suggestions. We feel that these changes greatly strengthened the manuscript.

To be fully transparent, we want to mention the following correction of an error in our manuscript. While revising the manuscript, we became aware that a recording had been erroneously included in our dataset despite it not meeting the criteria for inclusion (we impose a cut-off time of 5 ms from the onset of the stimulation to the onset of the IPSC. This cell was well outside that range at 9 ms). The removal of this single data point has altered several figure panels slightly but has not otherwise impacted the overall findings from the paper.

Reviewer #1 (Remarks to the Author):

This manuscript from Alanna Watt and her group shows that neurons in the cerebellar nuclei (CN neurons) may receive input from Purkinje cells located in different cerebellar lobules. This seems to be the case in roughly 50 percent of target neurons, while the remaining neurons get their input from Purkinje cells located in one lobule only. The study provides a connectivity map that is based on optogenetic activation of Purkinje cell axons and electrophysiological recordings from CN neurons in slice. Finally, the authors investigate how much current is needed to affect spike firing in CN neurons and come up with a low number of about 60pA. This work is very interesting and has the potential to have a very high impact on the field. As always for papers from this lab, the work is excellent in that it is well documented, beautifully illustrated and just a joy to read. There are two main comments that I have about the core claims that are made here:

We thank the reviewer for their positive comments and feedback regarding our results and data visualization, as well as for their careful reading of our manuscript. We have addressed the reviewer's concerns below.

1) The key message of the manuscript is that Purkinje cells from different cerebellar lobules can converge on the same target neurons in the cerebellar nuclei. If true, this finding will have substantial consequences on how we see computations at the output stage of the cerebellum arising. In that light, it is somewhat surprising that the authors do not offer controls that would exclude the possibility that blue light activation accidentally co-activates axons arising from other lobules. I did not find any control or comment about this, other than a reference to a Frontiers paper, but when looking that up, it also does not address the issue. However likely or unlikely this accidental co-activation is about to happen, a control needs to be provided.

There might be different ways to do this. I could imagine one of these two working, but others might work as well:

a) if optogenetic axonal activation elicits antidromic spikes that can be measured in Purkinje cell somata, the authors could test whether Purkinje cells in adjacent lobules ever get activated by axonal stimulation in the neighbor lobule. If so, this would mean that by axonal stimulation, the authors were not truly able to separate Purkinje cell output by lobule. And

b) if such antidromic spikes are not seen and thus cannot be used as a tool here, the authors could pick up on an observation from their *Frontiers* paper. A threshold duration of the blue light pulse is needed for activation. If there is unintended cross-talk of optogenetic activation, applying a subthreshold duration light pulse to axon bundle A and another to B (sequentially) would still lead to CN neuron activation, because of the cross-talk between A and B. There might be other tests to probe for unintended cross-activation, but a control must be provided.

We thank the reviewer for highlighting this important issue. To provide an additional control, we have optically stimulated while patching a Purkinje cell, as we are able to elicit antidromic spikes from the same lobule but not from neighbouring lobules (*option a* outlined above). This new data is included in new Supplementary Fig. 1. We observed antidromic spikes in every patched Purkinje cell when stimulating the same lobule. However, we observed no spikes when stimulating neighbouring, or adjacent, lobules other than in a single case where we recorded a single antidromic spike, with no spikes produced in other waves. The timing of the spike was slow but was just within the detection window for spikes in our dataset (5 ms). This spike may represent a rare spontaneous spike that is not time-locked to the stimulation, which occasionally occur in both optogenetic and extracellular stimulation experiments. However, even in the case that this does represent a light-evoked spike in a neighbouring lobule, it is a rare occurrence.

Further support of this is provided by our data where the high proportion of CN neurons receive single-lobule input in our dataset (> 50% of CN neurons that respond to lobule photostimulation). Taken together, our new control and our lobule stimulation dataset suggest that accidental cross-activation by optogenetic stimulation is not likely playing a role in the convergence patterns we present.

2) Fig 6: the conclusion from the data presented that an unexpectedly small amount of current is sufficient to affect spike firing is a bit puzzling. First, earlier recordings from DCN neurons have shown that starting with about 100pA currents, these neurons fire rebound spikes (Aizenman / Linden). Not the same experiment, but it also shows that currents in the same order of magnitude alter spike firing in these neurons. That means this observation – at the level of slice recordings – is neither surprising nor new. Second, it is not clear to me how informative this finding is. An (artificial) light pulse or current injection of 60pA has a different effect than, for example, a Purkinje cell complex spike, for a multitude of reasons, including

the very different kinetics. Finally, the question is whether this value would be different when recorded in vivo. For these reasons, the authors should provide more appropriate context when discussing their finding, and they should be open to discuss the technical limitations here.

We thank the reviewer for offering this insight highlighting this concern. To address this, we rewrote the rationale in the Results and the Discussion sections. We have removed mention of this as a surprising finding and we have better situated it in the context of experiments demonstrating Purkinje cell IPSP effects on CN neuron rebound depolarization in the Discussion section of our revised manuscript.

Reviewer #2 (Remarks to the Author):

In this MS, the authors recorded neurons in the medial cerebellar nucleus in brain slices while stimulating Purkinje cell axons at the bottom of individual cerebellar lobules using optogenetic. They found that single cerebellar neuron often receives Purkinje cell inputs from different lobules belonging to different antero-posterior zones. Connectivity pattern were found non-random, but no clear relationships between connectivity patterns and functional or morphological features were observed. Finally, the activation of Purkinje cells in one lobule is enough to induce a pause in nuclear neurons.

Overall, this MS is interesting as it addresses functional cortico-nuclear connectivity in brain slices using optogenetic stimulation of Purkinje cells in individual lobule. The authors should be commended for this original way to assess cerebellar physiology. However, technical limitations associated to the poor contextualization of the data in the literature of cerebellar modules limit the scope of the MS

We thank the reviewer for their positive comment and careful reading of our manuscript, as well as for their helpful feedback and suggestions. We have addressed the reviewer's concerns below and in the manuscript.

Main points :

-In this MS, the authors emphasized the antero-posterior zonation of the cerebellum, with gross anatomical distinction such as anterior, central, posterior and nodular. These zones have certainly a developmental and physiological relevance as they were associated with specific task and behaviors both in rodents and humans (from being motor in the anterior to oculomotor in the central and non motor in the posterior parts). However, since decades now, the functional zones of the cerebellum are defined longitudinally (i.e. in the medio-lateral axis) as a combination of olivo-cortical and corticonuclear projections that define cerebellar modules and more precisely microzones. Cerebellar modules encompass several lobules,

which are not necessarily contiguous (e.g. lobule III-IV with lobule VIII, Pijpers et al J Neurosci 2008, Ruigrok et al EJM 2008). Assumption from line 86: “we hypothesized that Purkinje cells in lobules belonging to the same functional zone were more likely to converge on a CN neuron than Purkinje cells in lobules from across zones” is not fully relevant based on the literature. Jan Voogd, Tom Ruigrok, Carl-Fredrik Ekerot, Henrik Jorntell, Richard Apps, Richard Hawkes, Izumi Sugihara and many other laboratories have spent years to study and refine the notion of anatomo-functional units in the cerebellum. In that respect, the multi-lobular Purkinje cell projection to cerebellar nuclei has been shown already several times (Voogd and Ruigrok, 2004, their Fig 4. Also shown by the group of Sugihara) using anatomical tracers. Actually, based on the maps from Sugihara’s lab (J Neurosci 2004 and 2007) or from Ruigrok’s lab (Ruigrok et al, EJM 2008), one could make precise predictions on PC convergence in cerebellar nuclei. Since the authors rather confirm anatomical predictions, at least an exhaustive comparison with the literature is necessary.

We thank the reviewer for this suggestion. We have included a more extensive discussion of the anatomical literature in our updated manuscript. We agree that the mediolateral (longitudinal) zonation of cerebellar modules no doubt plays a crucial role in Purkinje cell – CN neuron convergence. However, here we wanted to further inspect convergence by evaluating whether there might be additional patterning of Purkinje cell – CN neuron convergence within the sagittal plane. While a great deal of anatomical work has been done on this question, functional connectivity within the cerebellar circuit has not been as well studied. We feel that our work is a complement to anatomical studies, and rather than contradicting this work, it extends our understanding of cerebellar cortical output. We have provided more context throughout the manuscript including in the Introduction, Results and Discussion sections of our revised manuscript to address the previous anatomical literature more clearly.

-As an extension of the first point, the vermis is composed of several sagittal zones: A1, Ax, and B from midline to lateral areas. Each of these zones have specific nuclear projections. For example, the lateral parts of A1 and B zones do not project to the medial nucleus but to the lateral vestibular nucleus (LVN) outside the cerebellum. Ax has also specific projections. Therefore, in the sagittal slices used in this MS, it is important to identify their medio-lateral position. A1+B zones are > 600 μm wide, which means at least 3 slices from this study. Mediolateral information would help making sense of the distribution observed.

We thank the reviewer for raising this important point. We have modified our manuscript to clarify the relationship between mediolateral position and CN neuron distribution. We recorded from neurons within three neighbouring slices containing the fastigial CN, totaling a width of 600 μm (see new Supplementary Fig. 13). The fastigial CN in these slices correspond to areas in Groups I, II, and III from subregions of which are known to receive input from cerebellar cortical zones A, A1, and AX (Sugihara and Shinoda, 2007). We observed Purkinje cell input from multiple transverse zones in each of these three slices representing different positions along the mediolateral

axis of the cerebellum. Since individual microzones can vary in width between 100 to 300 μm (Apps and Hawkes, 2009), our study likely describes Purkinje cell input from a few microzones, and thus likely only reflect a few distinct cortico-nucleo-olivary loops. We have tried to better represent the distribution of cells in the new Supplementary Fig. 11 to aid in interpretation.

specific points:

-If I understand correctly, there are 3/71 (3/41 connected ones) neurons receiving Purkinje cell inputs from 4 zones, which means around 5%. I am afraid that they cannot be considered as an important group as is.

While we agree that CN neurons receiving input from all 4 transverse zones is not a large proportion of the neurons we have recorded (although our data represent a lower, not upper bound of their prevalence in the nucleus), our modeling shows that they are observed at much higher frequencies than expected by chance. Whether they are important or not remains to be determined. Nonetheless, they represent a newly-identified site where multi-modal input can converge in the cerebellum.

There is abundant evidence in the nervous system that small number of neurons can have a disproportionate influence on the function of the nervous system. Classic examples are hub neurons in the developing hippocampus (Bonifazi et al., 2009) and trigger neurons in the neocortex (Kozloski et al., 2001). We have added a section to the Discussion to address the potential role(s) of these neurons in more detail.

-I don't fully understand the rationale of the hierarchical clustering of Fig2b. It seems that lobule VI/VII is distant from others essentially because half of the neurons in this group were only connected to this lobule. Also, if we exclude the first half of the plot, in which neurons are connected to a single lobule, in the other half, they are connected to different antero-posterior zones. Typically, lobule VI/VII and at least one other. Hence, conclusion stated line 91 does not seem to be supported by the figure. Maybe machine learning algorithms (non-supervised or supervised) using several parameters as features (identity and number of connected lobules, position in the CN, firing rate....) could help isolating specific clusters of neurons even if simple correlations (Figure 3) could not.

We thank the reviewer for raising this issue. We chose to use an unsupervised hierarchical clustering algorithm to analyze this data without any biases, and thus started on our full data set including CN neurons receiving innervation from a single lobule. The fact that our data clustered into the same transverse zones that have been reported in the past using different methods was interesting to us. It suggests that the same functional divisions that are observed at the input level to the cerebellar cortex are respected at the output level as well. We have rewritten the Discussion to highlight this more clearly.

We considered using machine learning algorithms to analyze our data in more detail but given the low-yield nature of these experiments and the large sample size required for machine learning, we were unable to combine the many parameters we evaluated to determine clusters of CN neurons from the cells we recorded.

As to the observation regarding lobule VI/VII being connected to many other lobules. Lobule VI/VII is the sole lobule (despite it being a combination of two fused lobules) in the central zone, meaning that experimentally, we stimulated this zone once with a single light pulse at the base of the lobule. Thus, we are unable to compare whether lobule VI/VII would be more often connected to another central zone lobule, since we cannot experimentally make this comparison. Thus, we cannot compare the inter-connectedness of this zone with another one, for example the anterior zone where we stimulated three lobules, II, III, and IV/V, within the zone.

-Most of the neurons seem located in the caudo-dorsal part of the medial nucleus, does it explain why half of them receive inputs from lobule the central zone?

We thank the Reviewer for highlighting this point. Please see our response to Reviewer #3 below.

-Why recordings were made at -60 mV in potassium gluconate internal solution, which is then very close of the chloride reversal potential if junction potential is added (around -10 mV more)? Could connected sites be missed?

We chose to record from CN neurons at -60 mV using potassium gluconate internal solution to maintain near-physiological conditions for our experiments. Many researchers have chosen to study Purkinje cell – CN neuron synapses using high-chloride internal solutions and/or voltage clamping to 0 mV to ensure that small synapses were not missed. However, our experimental design required us to make long recordings from CN neurons (~20-30 minutes) to sample connectivity from multiple regions of the cerebellar cortex. We thus chose to use a physiological synaptic approach to ensure we could record from CN neurons for as long as possible. Although the smallest synaptic connections observed in our experiments are ~12 pA, which gives us confidence that we are able to sample a wide range of amplitudes of synaptic inputs, it is nonetheless possible that the smallest synaptic inputs are missed.

-In previous studies, it was shown that the decay time constant of the GABAergic current from Purkinje cells to principal cells in the nuclei was ultra fast (Person and Raman, Nature 2011), while much slower at the Purkinje cell to nucleo-olivary neurons (Najac and Raman). Could the author evaluate the dataset in function of the decay time constant? Also, excitatory vs inhibitory neurons can be distinguished by their firing rates and spike shape (Ozcan et al, J Neurosci 2020 and Uusisaari et al J Neurophysiol 2007), is it possible to sort them by this feature?

We thank the reviewer for these insightful suggestions. The decay time constant of GABAergic currents on CN neurons appears to differ based on the particular nucleus sampled during experiments. While the recordings described in both Person and Raman (2012) and Najac and Raman (2015) were performed in the interposed and lateral (dentate) nuclei, those performed in the fastigial (medial) CN also by the Raman lab (Vaaga et al., 2020) showed similar decay time constants (~6-7 ms) to those we observed in our data (Supplementary Fig. 2). The decay time constants that both Vaaga & Raman (Vaaga et al., 2020) we describe here are also much faster than those observed in nucleo-olivary CN neurons (~40 ms) (Najac and Raman, 2015); similar to those seen in Uusisaari & Knöpfel (2008), suggesting that the CN neuron population described in our manuscript likely reflects a population of predominantly excitatory CN neurons. Additionally, while excitatory and inhibitory neurons have been shown to exhibit differences in their firing rates and spike shapes, these differences tend to be quite subtle, where there is much overlap in the distributions of both firing rates and spike width of excitatory and inhibitory CN neurons (Uusisaari et al., 2007), particularly in slice recordings made at room temperature. *In vivo* recordings from excitatory (glutamatergic) and inhibitory (GABAergic) CN neurons have shown that excitatory CN neurons exhibit a fast firing rate (mean ~65 Hz) compared to GABAergic CN neurons (~25 Hz) (Ozcan et al., 2020).

We have provided a new Supplementary Figure 5 depicting the position of CN neurons in the fastigial CN based on their firing rates and number of input zones. We observed 23 of 41 cells with identified connections that we recorded spontaneous firing in cell-attached mode before breaking through into whole-cell mode, so we were unable to assess the spike shape using current clamp. Since differences in firing rate between excitatory and inhibitory cells are overlapping, we could not distinguish between putative excitatory and inhibitory CN neurons.

Reviewer #3 (Remarks to the Author):

Gruver et al have submitted a manuscript entitled, "Structured connectivity in the output of the cerebellar cortex." This paper seeks to evaluate a long held question of whether cerebellar nuclei neurons integrate information from multiple regions of the cerebellum. The authors perform a rigorous set of experiments – using slice physiology to examine Purkinje cell inputs into the cerebellar nucleus.

It is known that signals from multiple Purkinje cells must be integrated at the level of cerebellar nuclear neurons because the number of Purkinje cells is >50x times greater than that of cerebellar nuclear neurons. However, the anatomical organization of this Purkinje cell convergence has not been well described – with numerous outstanding possibilities including but not limited to whether nearby PCs make convergent inputs, whether PCs belonging to the same module [Ito, 1984] make convergent inputs, or whether other forms of integration exist including connections from diverse regions across the cerebellum. These fundamental

questions could shed light on cerebellar function and how cerebellar nuclei process information from the cerebellar cortex.

To get at this question, the authors combine slice physiology with focal optogenetic stimulation in vermal slices. They demonstrate that spatially segregated Purkinje neurons in different cerebellar lobules can make convergent inputs into a single fastigial/medial nuclear neuron and raises the possibility that previous findings across multiple species pointing to duplicated or triplicated functional topography within the cerebellum might be mediated by convergent Purkinje to cerebellar output neurons. These findings are important and are performed for the most part in rigorous fashion.

We thank Reviewer #3 for their positive comments and helpful feedback.

Several points, however, should be raised. Although these findings are important and could reveal fundamental insight into cerebellar function, the paper remains limited to this finding. The authors show the existence of convergence but do not provide functional insights into the function of these findings. Many potential questions could be addressed including but definitely not limited to: is the convergence necessary? Is the convergence redundant, what is the functional significance of this convergence, how generalizable is this finding outside of the limited subset of cerebellar nuclear neurons evaluated. Although adding new experiments of comprehensive in vivo functional analysis and/or anatomical analysis could resolve the current limitations, the amount of work needed for any of these directions would be considerably large to perform in the timeframe for the revision.

We thank the reviewer for their insightful comments and questions that we ourselves have also asked in light of our findings. It is our view that identifying patterns of convergence between Purkinje cells and CN neurons in the fastigial CN is an important step in the direction of understanding cerebellar information processing and output as a whole. To understand what this convergence does is outside the scope of this paper, however. We had added to the Discussion to highlight these important future directions.

Specific comments.

1. The authors conclude that there are distinct fastigial neurons that (1) receive inputs from a single zone and (2) receive inputs from multiple zones. However, because of the limitation of the slice thickness being 200 um, the true existence of (1) as well as subdivisions within (2), remain in question. To their credit, the authors address this limitation in the discussion and note that this may limit the conclusions that can be drawn from this study despite still making the very conclusion that is made. The organization of the current manuscript, then, could be clarified to minimize confusion and to prevent any overstatement of manuscript findings.

We thank the reviewer for these comments. We have included a new schematic in Figure 1 to highlight that these experiments are performed in slices, and we have included this information in the Abstract as well as throughout the manuscript to ensure that we do not overstate the manuscript findings.

2. It is also important to note the cerebellar nuclear neurons that were recorded are a very small group of neurons localized to a very specific caudal region within the medial nuclei (figure 4, supp figure 6). This fact makes it difficult to address whether the findings made are specific to this population. Also, this region is known to be comprised of morphologically similar neurons (Golgi staining analysis: Beitz and Chan-Palay, 1979, Neuroscience), which could contribute to the morphologic similarities noted by the authors. Importantly, the authors also suggest that CN neurons receiving diverse input are biased to caudal fastigial regions. However, with the specific sampling done, it's also quite possible that sampling bias could contribute to these findings. In fact, it is known that the rostral fastigial also receive inputs as a population from multiple vermal lobules (Sugihara et al 2009); not having these regions tested could thus impact the strength of conclusions that can be made. In conjunction with the concerns raised above, it raises concern about generalizability and/or the extent of conclusions that can be made.

We thank the reviewer for their careful reading of the paper. We randomly chose CN neurons to record from based on visual properties of the cell in the slice. Unexpectedly, we appear to have been more successful patching neurons in the caudo-dorsal part of the nucleus than in the rostro-ventral part of the nucleus. To clarify our sampling strategy: in each slice used, we performed the same line-by-line scanning process, moving from ventral to dorsal, to visually identify CN neurons. While we acknowledge that some bias in which cells we have selected seems to exist (which we address below), we also point out that due to the 3-dimensional shape of the fastigial nucleus, there is simply more nucleus in the dorsal region than in the ventral region (see below, image taken from Fig. 4g).

The apparent sampling bias favoring CN neurons in the dorsal region of the fastigial nucleus was unintentional; we attempted to record from CN neurons equitably throughout the fastigial CN but

were less successful at recording from neurons located in the rostroventral area of the fastigial CN, which appeared less optimal for patching. These rostroventral CN neurons likely belong to Group III as described by Sugihara and Shinoda (2004) and the F1R region recently described by Fujita et al. (Fujita et al., 2020), including the largest glutamatergic CN projection neurons (Fujita et al., 2020) as well as large glycinergic projection neurons (Bagnall et al., 2009).

We hypothesize three factors that may contribute to this uneven sampling. **1.** The large putative-glutamatergic projection neurons in the CN are thickly wrapped by perineuronal nets (Carulli et al., 2006), making them harder to visualize and thus patch. **2.** Due to their ventral position within the nucleus, these cells may be surrounded by a higher density of axons entering and leaving the cerebellum, impeding our ability to successfully visualize and target these cells for recording. **3.** Since our blade always entered the slice at the most dorsal area of the cerebellum during slicing, it is possible that tissue located most ventrally (at the bottom of each acquired slice) may be subject to more damage than the more dorsal areas of the fastigial CN that encounter the slicing blade earlier. For these reasons, we argue that this bias is unavoidable in slice recordings.

We have added more context to the Methods section to describe our CN neuron sampling process. Further, we have produced a new Supplementary Figure 13 to better demonstrate the spatial distribution of CN neurons we successfully patched. We have added text in the Results and the Discussion to address these concerns.

3. The authors note that their findings suggest that “widespread Purkinje cell synchrony may not be necessary to influence cerebellar output”, by showing that even small input stimulation can induce pauses in firing. However, the authors to my knowledge do not show what that small input/stimulation is actually resulting in in terms of Purkinje neuronal stimulation. Synchrony as determined by the referenced study by Person and Raman 2011 show that even a small amount of synchrony – even 2 afferents – can be sufficient. Thus, it is not clear whether these data are sufficient to support this claim.

We thank the reviewer for this important suggestion. We have edited the Abstract, the Results and Discussion to ensure that we are not overstating the implications of our data.

4. As alluded to above, these data are not put into a functional or anatomical context either of which would dramatically affect the impact of these data. The AZ, CZ, PZ, and NZ are the classifications originally developed to indicate that these multiple zones harbor different Purkinje subtypes (Ozol et al., 1999, J Comp Neurol; Sillitoe and Joyner, 2007, Annu Rev Cell Dev Biol). At a circuit level, it is hypothesized that Purkinje neurons that make convergent input to the CN are located within the same cerebellar stripes (Sugihara et al., 2009, J Comp Neurol; Fujita and Sugihara, 2013, Front Neural Circuits). How the authors’ findings map on to this modular organization within the cerebellum is not clear from the manuscript.

We thank the reviewer for highlighting this point. We have produced a new Supplementary Figure 13 to demonstrate the relationship between our observed Purkinje cell – CN connectivity patterns and the mediolateral organization of the cerebellar vermis. To summarize what we described in a previous response, we recorded from 3 continuous slices for a total width of 600 μm . The fastigial CN in these slices corresponds to areas in Groups I, II, and III from Sugihara & Shinoda (2007) which are known to receive input from cerebellar cortical zones A, A1, and AX. We observed Purkinje cell input from a variety of transverse zones in each of these 3 slices representing different positions along the mediolateral axis of the cerebellum.

Additionally, we attempted to immunolabel post-fixed slices containing patched cells after electrophysiology experiments with a marker for zebrin II, but we encountered prohibitive signal-to-noise ratio issues when performing immunohistochemistry experiments with this non-perfused tissue. For this reason, our analysis focused 3D localization of cells based on data collected during live experiments. Whether the convergence occurs between the same individual Purkinje cells located in these zones or varies along a given cerebellar stripe is an open question, but the observation of complex combinations of convergence patterns regardless of mediolateral position offers new insight into cerebellar information transfer that was previously uncharacterized.

5. The anatomical demonstration of convergence in Figure 5 could be more convincing. Since the dendrites of nuclear cells are densely packed, $\sim 20 \mu\text{m}$ proximity, as the authors show, the data do not necessarily indicate input to the cells of interest. Indeed, it's potentially feasible that signals are not necessarily of terminals but of axonal segments that may be passing through the plane.

We thank the reviewer for highlighting this area for improvement. To address this, we stained for the vesicular GABA transporter VGAT in a new subset of animals injected with a fluorescently-labeled AAV (pAAV-CAG-GFP) and evaluated how many GFP-positive Purkinje cell puncta were also positive for VGAT. We found that $> 40\%$ of these puncta express the VGAT label, suggesting that nearly half of puncta whose distances we compared in Fig. 5 likely represent Purkinje cell presynaptic terminals. Additionally, after measuring the size of VGAT-positive puncta, we adjusted the lower limit of our puncta area threshold value to better account for the size of potential synaptic terminals. We have re-analyzed our data included in Fig. 5 and updated the graph in Fig. 5e to reflect this.

References

- Apps R, Hawkes R (2009) Cerebellar cortical organization: a one-map hypothesis. *Nat Rev Neurosci* 10:670-681.
- Bagnall MW, Zingg B, Sakatos A, Moghadam SH, Zeilhofer HU, du Lac S (2009) Glycinergic projection neurons of the cerebellum. *J Neurosci* 29:10104-10110.

- Bonifazi P, Goldin M, Picardo MA, Jorquera I, Cattani A, Bianconi G, Represa A, Ben-Ari Y, Cossart R (2009) GABAergic hub neurons orchestrate synchrony in developing hippocampal networks. *Science* 326:1419-1424.
- Carulli D, Rhodes KE, Brown DJ, Bonnert TP, Pollack SJ, Oliver K, Strata P, Fawcett JW (2006) Composition of perineuronal nets in the adult rat cerebellum and the cellular origin of their components. *The Journal of comparative neurology* 494:559-577.
- Fujita H, Kodama T, du Lac S (2020) Modular output circuits of the fastigial nucleus for diverse motor and nonmotor functions of the cerebellar vermis. *Elife* 9.
- Kozloski J, Hamzei-Sichani F, Yuste R (2001) Stereotyped position of local synaptic targets in neocortex. *Science* 293:868-872.
- Najac M, Raman IM (2015) Integration of Purkinje cell inhibition by cerebellar nucleo-olivary neurons. *J Neurosci* 35:544-549.
- Ozcan OO, Wang X, Binda F, Dorgans K, De Zeeuw CI, Gao Z, Aertsen A, Kumar A, Isope P (2020) Differential Coding Strategies in Glutamatergic and GABAergic Neurons in the Medial Cerebellar Nucleus. *J Neurosci* 40:159-170.
- Person AL, Raman IM (2012) Purkinje neuron synchrony elicits time-locked spiking in the cerebellar nuclei. *Nature* 481:502-505.
- Sugihara I, Shinoda Y (2004) Molecular, topographic, and functional organization of the cerebellar cortex: a study with combined aldolase C and olivocerebellar labeling. *J Neurosci* 24:8771-8785.
- Sugihara I, Shinoda Y (2007) Molecular, topographic, and functional organization of the cerebellar nuclei: analysis by three-dimensional mapping of the olivonuclear projection and aldolase C labeling. *J Neurosci* 27:9696-9710.
- Uusisaari M, Knopfel T (2008) GABAergic synaptic communication in the GABAergic and non-GABAergic cells in the deep cerebellar nuclei. *Neuroscience* 156:537-549.
- Uusisaari M, Obata K, Knopfel T (2007) Morphological and electrophysiological properties of GABAergic and non-GABAergic cells in the deep cerebellar nuclei. *Journal of neurophysiology* 97:901-911.
- Vaaga CE, Brown ST, Raman IM (2020) Cerebellar modulation of synaptic input to freezing-related neurons in the periaqueductal gray. *Elife* 9.

Reviewers' Comments:

Reviewer #1:

Remarks to the Author:

All my previous concerns have been appropriately addressed.

Reviewer #2:

Remarks to the Author:

The authors have certainly improved the MS and most of the technical and conceptual limitations raised at the initial steps have been lifted.

Some concerns remain:

-precise cortico-nuclear connections have been anatomically previously described yielding clear hypotheses about multizonal structured connectivity (see Voogd, J., Ruigrok, T. J. H. 2004. *Journal of Neurocytology*, and Sugihara, I., Shinoda, Y. 2007. *The Journal of Neuroscience*). These articles are cited, but multizonal CN cells inputs should be clearly related to this anatomical multilobular organization previously described.

- The hierarchical clustering(HC) is still unclear to me. Could the author explain more in the methods what are the features and the individuals as in HC the dendrogram splits/groups individuals. Maybe showing the rough dendrogram would help.

I would also add one more point of discussion related to Purkinje cell synchronization. In the recording conditions at 34°C, Purkinje cells spike in slices, but it seems that few spontaneous currents were observed in individual traces (Fig 1d). The large light-evoked currents suggest that optogenetic stimulation at the base of the lobules likely synchronize many Purkinje cell axons.

So, it is clear that in fig 6, the effect on CN firing rate is the result of many synchronized inputs, even if they originate in a single lobule.

Response to Reviewers

We thank the reviewers for their comments, which we feel has greatly improved the manuscript. Below we outline our specific responses to their comments in **blue**.

REVIEWERS' COMMENTS

Reviewer #1 (Remarks to the Author):

All my previous concerns have been appropriately addressed.

Response: We thank the reviewer for their input on our manuscript.

Reviewer #2 (Remarks to the Author):

The authors have certainly improved the MS and most of the technical and conceptual limitations raised at the initial steps have been lifted.

Response: We thank the reviewer for their positive assessment of our revised manuscript.

Some concerns remain:

-precise cortico-nuclear connections have been anatomically previously described yielding clear hypotheses about multizonal structured connectivity (see Voogd, J., Ruigrok, T. J. H. 2004. Journal of Neurocytology, and Sugihara, I., Shinoda, Y. 2007. The Journal of Neuroscience). These articles are cited, but multizonal CN cells inputs should be clearly related to this anatomical multilobular organization previously described.

Response: We have added a section drawing a stronger link between our functional studies and the structural organization previously described on lines 239-241.

- The hierarchical clustering(HC) is still unclear to me. Could the author explain more in the methods what are the features and the individuals as in HC the dendrogram splits/groups individuals. Maybe showing the rough dendrogram would help.

Response: We have added an additional description of this in the Methods section on lines 404-410. The dendrogram is included at the top of Figure 2b.

I would also add one more point of discussion related to Purkinje cell synchronization. In the recording conditions at 34°C, Purkinje cells spike in slices, but it seems that few spontaneous currents were observed in individual traces (Fig 1d). The large light-evoked currents suggest that optogenetic stimulation at the base of the lobules likely synchronize many Purkinje cell axons.

So, it is clear that in fig 6, the effect on CN firing rate is the result of many synchronized inputs, even if they originate in a single lobule.

Response: We thank the reviewer for their insight. We have rewritten the text to better reflect this on lines 201-203.